# Viable fertilizer prescription model: Soil test crop response approach for sustained and targeted yield, quality of Coriander (*Coriandrum sativum L.*) in Alfisols

Krishna Murthy Rangaiah[1]*, Bhavya Nagaraju[2], Govinda Kasturappa[1], Shivakumara Maragondanadibba Nanjundappa[1], Sanjay Srivastava[3‡], Pradip Dey[4‡], Immanuel Chongboi Haokip[3‡]

**1** All India Coordinated Research Project on Soil Test Crop Response, University of Agricultural Sciences, Bangalore, India, **2** Department of Soil Science and Agricultural Chemistry, University of Agricultural Sciences, Bangalore, India, **3** ICAR-Indian Institute of Soil Science, Bhopal, India, **4** Indian Council of Agricultural Research-Agricultural Technology Application Research Institute, Kolkata, India

☯ These authors contributed equally to this work.
‡ SS, PD and ICH also contributed equally to this work.
* stcruasbangalore@gmail.com

## Abstract

A comprehensive three-year field study was conducted to develop and validate soil test crop response (STCR)–based nutrient prescription equations for coriander (Coriandrum sativum L.) in Alfisols. The research comprised three distinct experimental phases: a fertility gradient experiment, a main experiment, and a validation trial. Across these phases, over 72 soil and plant samples were collected and analyzed for nutrient content and compositional quality. The fertility gradient experiment established significant variability in soil NPK status using fodder maize as an indicator crop. In the main experiment, coriander plots subjected to varying fertility levels and nutrient management strategies—including farmyard manure (FYM) supplementation—were assessed for yield, nutrient uptake, and key quality metrics (total phenols, flavonoids, and ascorbic acid). Prescription equations targeting specific yield levels were developed by calculating nutrient requirements and quantifying contributions from soil, fertilizer, and FYM. The validation trial demonstrated that STCR-based recommendations, especially those integrating 7.5 t ha$^{-1}$ FYM, substantially improved green foliage yield (up to 56.9% higher), nutrient uptake (N, P, K), and quality parameters compared to generalized recommended doses and soil fertility rating approaches. Enhanced outcomes were observed for value cost ratio, response yield stick, and nutrient use efficiency indices (Partial Factor Productivity, Agronomic Efficiency, Partial Nutrient Balance, Internal Utilization Efficiency). Overall, the STCR approach with integrated nutrient management proved effective in increasing yield, improving quality, and optimizing nutrient use efficiency in coriander, advancing the case for site-specific and balanced fertilizer application in sustainable coriander production.

**Data availability statement:** All relevant data are within the manuscript and its Supporting information files.

**Funding:** The study was funded by the Indian Council of Agricultural Research and the University of Agricultural Sciences, Bangalore (Grant number: CRP-18). The funders contributed to study design, data collection, and analysis but did not provide funding for publication costs or manuscript preparation. The funders had no role in the decision to publish or in the preparation of the manuscript.

**Competing interests:** The author reported no potential conflict of interest.

**Abbreviations:** STCR, Soil Test Crop Response; SFR, Soil Fertility Rating; N, Nitrogen; P, Phosphorus; K, Potassium; NR, Nutrient Requirement; CS, Contribution from Soil; CF, Contribution from Fertilizer; CFYM, Contribution from Farmyard Manure; GRD, General Recommended Dose; RYS, Response Yard Stick; VCR, Value Cost Ratio; PFP, Partial Factor Productivity; AE, Agronomic Efficiency; PNB, Partial Nutrient Balance; IUE, Internal Utilization Efficiency; SOC, Soil Organic Carbon; FYM, Farmyard Manure; OM, Organic Manure; NPK, Combined Nitrogen, Phosphorus, and Potassium; TY, Target Yield; RBD, Randomized Block Design; CV, Coefficient of Variation; SD, Standard Deviation; ANOVA, Analysis of Variance; LSD, Least Significant Difference; DTPA, Diethylene Triamine Pentaacetic Acid; EC, Electrical Conductivity; CEC, Cation Exchange Capacity; SPSS, Statistical Package for Social Sciences; OPSTAT, Online Statistical Programme for Agriculture; MS, Microsoft, referring to Excel or Office applications.

## Introduction

Fertilizer recommendations are crucial in enhancing crop yield and quality [1]. Traditionally, fertilizer recommendations have often followed blanket approaches, where uniform doses are suggested across a region, disregarding the inherent variability in soil fertility [2]. Other approaches include soil test ratings with adjustments, nutrient removal-based recommendations, yield goal-based recommendations, and sufficiency level approaches. While these methods offer some level of guidance, they often fail to account for the specific nutrient-supplying capacity of individual fields and the precise nutrient requirements to achieve a targeted yield [3]. Alternative approaches, such as Variable Rate Techniques (VRT), utilize geophysical field zoning and site-specific management for optimizing nutrient application [4] The Soil Test Crop Response (STCR) approach emerges as a more refined and scientific methodology for fertilizer recommendations. Unlike blanket recommendations, STCR focuses on establishing a quantitative relationship between soil test values, crop response to applied nutrients, and the desired yield. This approach develops specific fertilizer prescription equations that consider the initial soil fertility status, the nutrient requirements for a particular yield target, and the efficiency of both soil-applied and fertilizer nutrients. The importance of the STCR approach lies in its ability to provide site-specific and need-based fertilizer recommendations. This not only optimizes nutrient use efficiency, leading to higher yields and improved profitability for farmers, but also minimizes the environmental risks associated with excessive or imbalanced fertilizer application [5–7].

Coriander (*Coriandrum sativum* L.), a widely grown condiment crop of the tropics is very popular in cuisines to food court. India is the largest producer of coriander both in terms of area (547421 ha) and production (527390 tonnes), with Madhya Pradesh, Gujarat, and Rajasthan being the top producers [8,9]. It is not only used for culinary purposes but also as a condiment as well as a nutraceutical. It is an herbaceous annual member of the Apiaceae family under the order Apiales. Coriander requires a cool climate during the vegetative growth stage and a warm, dry climate at flowering. It can be cultivated in almost all types of soils but in well-drained loamy soil it flourishes well [10].

The productivity of coriander is influenced by several factors such as soil, variety, fertilizer management, and various agro techniques used for growing the crop. Nutrients play a vital role in the functioning of normal physiological processes during the period of growth and development of plants [11]. However, for obtaining a higher economic yield, a balanced supply of nutrients is one of the key factors. Nitrogen is an essential nutrient in creating the plant dry matter, as well as many energy-rich compounds that regulate photosynthesis and plant production [12]. Phosphorus is essential for cell division and for development of meristematic tissues and it is very important for carbohydrate transformation due to multitude of phosphorylation reactions and to energy rich phosphate bond [13]. Potassium is important for growth and elongation probably due to its function as an osmotic and may react synergistically with IAA. Moreover, it promotes $CO_2$ assimilation and translation of carbohydrates from the leaves to storage tissues [14]. By integrating soil testing, plant nutrient

requirements at different growth stages, and yield targets, this study aims to formulate precise fertilizer recommendations that optimize nutrient uptake and utilization. The outcomes of this research are expected to provide valuable insights for improving coriander productivity, reducing fertilizer wastage, and promoting sustainable coriander cultivation practices in India. A growing body of research demonstrates that long-term nutrient supply options, particularly those improving soil phosphorus availability, are crucial for sustainability in intensive cropping systems [7].

## Materials and methods

### Details of the experimental field

Field experiments were conducted from 2022 to 2024 at the Zonal Agricultural Research Station, University of Agricultural Sciences, Bangalore, Karnataka, India (13° 04' 55.2'' N latitude, 77° 34' 10.0'' E longitude). The rainfall data reveals significant inter-annual and seasonal variability across the three years (2022–2024), with annual total ranging from 829.2 mm in 2023 to 1555.8 mm in 2022, indicating a 46.8% reduction during the drought year, followed by partial recovery to 1380.1 mm in 2024. The temporal distribution demonstrates a distinct monsoon pattern with peak precipitation typically occurring between May and October, though 2024 exhibited exceptional variability with the highest recorded monthly rainfall of 585.4 mm in October (S1 Table). The experimental site features Typic Kandic Paleustalfs, classified as a fine mixed Isohyperthermic family [15]. Surface soil samples (0–20 cm depth) were collected using standard sampling protocols and analyzed for physico-chemical properties following the procedures described by [16]. Particle size distribution was determined by the International Pipette method, confirming a well-drained deep red sandy loam texture with sand, silt, and clay percentages of 63.12%, 12.55%, and 24.33%, respectively. Soil pH and electrical conductivity (EC) were measured, revealing acidic conditions (pH 5.98) and non-saline status (EC 0.30 dS m$^{-1}$). The cation exchange capacity was medium (19 cmol (p$^+$) kg$^{-1}$). Baseline soil fertility indicated high organic carbon (35.37 g kg$^{-1}$), sufficient available nitrogen (290 kg ha$^{-1}$), high available phosphorus (80.47 kg ha$^{-1}$), and medium available potassium (190 kg ha$^{-1}$). The soil's phosphorus and potassium fixation capacities were determined to be 180 and 100 kg ha$^{-1}$, respectively, using equilibrium methods [17]. DTPA-extractable micronutrients (Fe, Mn, Zn, Cu) were within adequate levels. The research followed a three-phase protocol (illustrated in Fig 1) to develop an STCR-based fertilizer prescription model: Phase I involved establishing a fertility gradient using fodder maize (kharif 2022); Phase II comprised the main experiment with coriander (kharif 2023); and Phase III served as a validation trial with coriander (kharif 2024) to assess the accuracy of the derived targeted yield equations.

### Fertility gradient experiment (phase I)

Initially, a fertility gradient experiment was established using Fodder maize var. African tall. Artificial fertility variation was created by dividing the experimental area into three equal rectangular strips: Low Fertility Strip (LFS), Medium Fertility Strip (MFS), and High Fertility Strip (HFS) [18]. These strips received varying fertilizer applications of N, $P_2O_5$, and $K_2O$ ($N_0P_0K_0$; $N_1P_1K_1$; $N_2P_2K_2$) (S1 Fig). LFS served as the unfertilized control. MFS received the general recommended nitrogen dose for fodder maize, while phosphorus and potassium were applied based on the soil's fixation capacities (180 kg ha$^{-1}$ for $P_2O_5$ and 100 kg ha$^{-1}$ for $K_2O$). HFS received double the fertilizer rates applied to MFS. Calculated fertilizer amounts were applied and incorporated by ploughing to ensure uniform distribution within each strip. Subsequently, fodder maize was sown and irrigated. The intensive cultivation of fodder maize resulted in changes in soil fertility levels. Sixty days post-sowing, the fodder maize was harvested from each fertility strip. Twenty-four soil samples were collected from each strip both before and after harvest and analyzed for available nitrogen ($KMnO_4$-N), phosphorus (Bray-$P_2O_5$), and potassium ($NH_4OAc$-$K_2O$). Post-harvest soil test values confirmed the successful development of the intended fertility gradient.

### Test crop experiment (phase II)

Following the establishment of the fertility gradient, a test crop experiment with coriander (Variety: Arka Isha) was conducted. After the fodder maize harvest, each of the three fertility strips (LFS, MFS, HFS) was further divided into 24

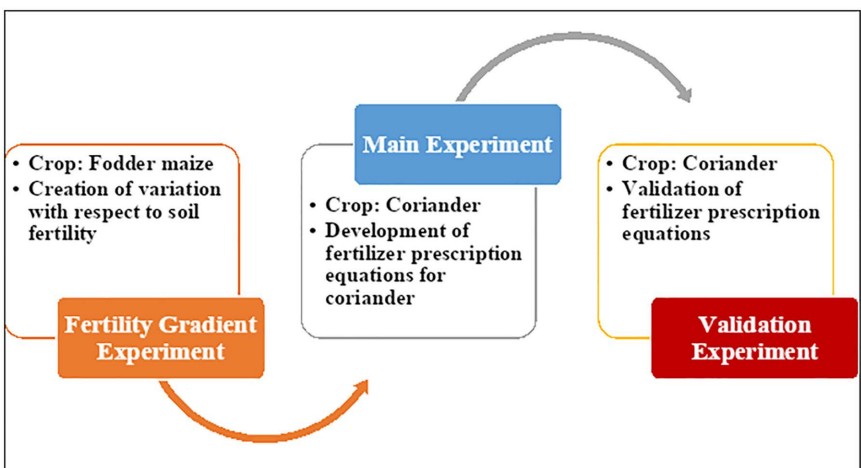

**Fig 1. Graphical representation of the experiment for the development of fertilizer prescription equation.**

individual plots, and soil samples were collected from each plot for NPK analysis. The main experiment with coriander comprised 24 treatments, involving four levels each of nitrogen (0, 17.5, 35, and 52.5 kg ha⁻¹), phosphorus (P₂O₅: 0, 17.5, 35, and 52.5 kg ha⁻¹), and potassium (K₂O: 0, 17.5, 35, and 52.5 kg ha⁻¹), and three levels of farmyard manure (FYM: 0, 7.5, and 15 t ha⁻¹). Within the experimental field, each fertility strip was subdivided into three sub-strips to accommodate the three FYM levels across the established gradient. The layout of this experiment, illustrating the major nutrient combinations for each treatment and the superimposed manure levels, is shown in S2 Fig. Coriander was cultivated using 21 selected treatment combinations and three control treatments, as detailed in Table 1. Care was taken to systematically impose NPK alone, NPK + FYM (7.5 t ha⁻¹), and NPK + FYM (15 t ha⁻¹) treatments across all fertility strips. The 21 fertilizer treatments and three control treatments were randomized within each strip to ensure that all 24 treatments were present in all three strips. All nutrient management and plant protection practices adhered to the package of practices of the University of Agricultural Sciences, Bangalore. FYM was applied 15 days before sowing. Basal applications included the entire P₂O₅, K₂O, and FYM, while nitrogen was applied in two splits: 50% at basal and 50% 25 days after sowing. Coriander green foliage is was harvested at 45 days after sowing, when the plants were about 6 inches (15 cm) tall. Plant samples were collected from each plot, processed, and analyzed for total NPK content. Nutrient uptake (NPK) by coriander was then calculated based on dry matter yield and the determined nutrient concentrations.

## Computation of basic parameters

Making use of the data on yield, nutrient uptake, pre sowing soil available nutrients, and applied fertilizer doses, the basic parameters viz., nutrient requirement (NR) and contributions of nutrients from soil (CS) and contributions of nutrients from fertilizers (CF) and contributions of nutrients from Farm Yard Manure (CFYM) were estimated [18,19]. By using the basic parameters, required doses of fertilizer N, P₂O₅ and K₂O were calculated following the STCR fertilizer prescription calculation technique under sole-fertilizer (NPK alone) and integrated (NPK + FYM) nutrient as follows [3].

### i) Fertilizer N, P and K alone

$$FN = \left( \frac{NR}{CF} \times 100 \times T \right) - \left( \frac{CS}{CF} \times SN \right)$$

**Table 1. Treatment structure for the main test crop experiment of coriander.**

| Sl. No. | N | P₂O₅ | K₂O | N (kg ha⁻¹) | P₂O₅ (kg ha⁻¹) | K₂O (kg ha⁻¹) |
|---|---|---|---|---|---|---|
| 1 | 0 | 0 | 0 | 0.0 | 0.0 | 0.0 |
| 2 | 0 | 0 | 0 | 0.0 | 0.0 | 0.0 |
| 3 | 0 | 0 | 0 | 0.0 | 0.0 | 0.0 |
| 4 | 0 | 2 | 2 | 0.0 | 35.0 | 35.0 |
| 5 | 1 | 1 | 1 | 17.5 | 17.5 | 17.5 |
| 6 | 1 | 1 | 2 | 17.5 | 17.5 | 35.0 |
| 7 | 1 | 2 | 1 | 17.5 | 35.0 | 17.5 |
| 8 | 1 | 2 | 2 | 17.5 | 35.0 | 35.0 |
| 9 | 2 | 0 | 2 | 35.0 | 0.0 | 35.0 |
| 10 | 2 | 1 | 1 | 35.0 | 17.5 | 17.5 |
| 11 | 2 | 1 | 2 | 35.0 | 17.5 | 35.0 |
| 12 | 2 | 2 | 0 | 35.0 | 35.0 | 0.0 |
| 13 | 2 | 2 | 1 | 35.0 | 35.0 | 17.5 |
| 14 | 2 | 2 | 3 | 35.0 | 35.0 | 52.5 |
| 15 | 2 | 2 | 2 | 35.0 | 35.0 | 35.0 |
| 16 | 2 | 3 | 2 | 35.0 | 52.5 | 35.0 |
| 17 | 2 | 3 | 3 | 35.0 | 52.5 | 52.5 |
| 18 | 3 | 1 | 1 | 52.5 | 17.5 | 17.5 |
| 19 | 3 | 2 | 2 | 52.5 | 35.0 | 35.0 |
| 20 | 3 | 2 | 1 | 52.5 | 35.0 | 17.5 |
| 21 | 3 | 2 | 3 | 52.5 | 35.0 | 52.5 |
| 22 | 3 | 3 | 1 | 52.5 | 52.5 | 17.5 |
| 23 | 3 | 3 | 2 | 52.5 | 52.5 | 35.0 |
| 24 | 3 | 3 | 3 | 52.5 | 52.5 | 52.5 |

$$FP = \left( \frac{NR}{CF} \times 100 \times T \right) - \left( \frac{CS}{CF} \times SP \right)$$

$$FK = \left( \frac{NR}{CF} \times 100 \times T \right) - \left( \frac{CS}{CF} \times SK \right)$$

**ii) Fertilizer N, P and K along with FYM**

$$FN = \left( \frac{NR}{CF} \times 100 \times T \right) - \left( \frac{CS}{CF} \times SN \right) - \left( \frac{CFYM}{CF} \times OM \right)$$

$$FP = \left( \frac{NR}{CF} \times 100 \times T \right) - \left( \frac{CS}{CF} \times SP \right) - \left( \frac{CFYM}{CF} \times OM \right)$$

$$FK = \left( \frac{NR}{CF} \times 100 \times T \right) - \left( \frac{CS}{CF} \times SK \right) - \left( \frac{CFYM}{CF} \times OM \right)$$

Where, T: Targeted yield (t ha$^{-1}$); FN: Quantity of nitrogen to be added through fertilizers (kg ha$^{-1}$); FP: Quantity of phosphorus to be added through fertilizers (kg ha$^{-1}$) FK: Quantity of potassium to be added through fertilizers (kg ha$^{-1}$) SN: Soil test value for available N (kg ha$^{-1}$); SP: Soil test value for available P (kg ha$^{-1}$) SK: Soil test value for available K (kg ha$^{-1}$); CFYM: Contribution of nutrients from FYM; OM: Quantity of FYM applied (t ha$^{-1}$).

## Validation experiment (phase III)

Finally, a validation field trial was conducted on the same soil series to verify the developed fertilizer prescription model for coriander (Variety: Arka Isha). Based on potential yield surveys, two coriander yield targets were selected: 10.00 t ha$^{-1}$ and 8.00 t ha$^{-1}$. This validation phase involved evaluating the percentage of achieved yield against the set targets, green foliage yield, response yield stick (RYS), economic viability through value cost ratio (VCR), and comparisons of nutrient use efficiencies with other fertilizer recommendation methods, namely the soil fertility rating (SFR) approach and the general recommended dose (GRD). The experiment was laid out in a randomized block design (RBD) with three replications. The treatments comprised: $T_1$ – STCR targeting 10 t ha$^{-1}$ (NPK alone); $T_2$ – STCR targeting 10 t ha$^{-1}$ (NPK+FYM); $T_3$ – STCR targeting 8 t ha$^{-1}$ (NPK alone); $T_4$ – STCR targeting 8 t ha$^{-1}$ (NPK+FYM); $T_5$ – General recommended dose; $T_6$ – Soil fertility rating; and $T_7$ – Absolute control. Before initiating the experiment, composite soil samples were collected from each plot at a depth of 0–20 cm, following the experimental layout. Fertilizer application for the STCR treatments ($T_1$ to $T_4$) was calculated using the derived STCR equations. The coriander crop was cultivated following standard practices outlined in the crop production guide, harvested at full maturity, and yields were calculated based on the net plot area, expressed in tons per hectare (t ha$^{-1}$). The Response yardstick (RYS), percent deviation, and Value Cost Ratio (VCR) were computed by using the standard formulae [18].

$$RYS = \frac{\text{Yield response } \left(\text{kg ha}^{-1}\right)}{\text{Total nutrient applied (kg ha}^{-1})}$$

$$\text{Percent deviation} = \frac{\left[\text{Actual yield obtained } \left(\text{kg ha}^{-1}\right) - \text{Targeted yield } \left(\text{kg ha}^{-1}\right)\right]}{\text{Targeted yield (kg ha}^{-1})} \times 100$$

$$VCR = \frac{\left[\text{Yield in treated plot } \left(\text{kg ha}^{-1}\right) - \text{Yield in control plot } \left(\text{kg ha}^{-1}\right)\right]}{\text{Cost of fertilizers and FYM applied to treated plot}} \times \text{Cost kg}^{-1} \text{ of leaf}$$

## Chemical analysis of soil and plant samples

In the study, a total of 330 soil samples and 186 plant samples were collected for chemical analysis throughout the study. Specifically, 72 soil samples were obtained from three fertility strips during the 2022 fertility gradient experiment, 72 soil samples were collected during the 2023 main experiment, and 42 composite soil samples along with corresponding plant samples were gathered during the 2024 validation experiment at both initiation and crop maturity. Consistently across all experimental phases, soil sampling was performed at pre-sowing and post-harvest stages, while plant samples were collected at crop harvest following standard analytical protocols.Soil samples were air-dried and then ground to pass through a 2 mm sieve. Table 1 and 2 summarizes the methods and references used for determining the availability of major nutrients in soil and plant samples, respectively. Nutrient uptake for each element was calculated based on the chemical analysis data and the dry matter yield of the plants.

**Table 2. Descriptive statistics of yield and nutrient uptake under the main experiment of coriander.**

| Strips | | Green foliage yield (t ha$^{-1}$) | Total N uptake (kg ha$^{-1}$) | Total P$_2$O$_5$ uptake (kg ha$^{-1}$) | Total K$_2$O uptake (kg ha$^{-1}$) |
|---|---|---|---|---|---|
| LFS | Range | 2.26-9.95 | 12.79-66.20 | 5.24-23.32 | 13.49-96.10 |
| | Mean±SD | 6.30±2.51 | 41.49±16.68 | 15.43±5.70 | 51.44±26.45 |
| | (CV %) | 39.84 | 40.20 | 36.94 | 51.42 |
| MFS | Range | 2.77-12.37 | 16.24-85.08 | 5.20- 53.56 | 23.98-104.04 |
| | Mean±SD | 6.99±2.74 | 44.50±19.48 | 19.18±10.04 | 63.20±23.86 |
| | (CV %) | 39.19 | 43.78 | 52.35 | 38.36 |
| HFS | Range | 3.36-11.36 | 21.13-71.18 | 10.09-30.40 | 29.17-104.62 |
| | Mean±SD | 7.57±2.25 | 49.22±17.74 | 22.56±5.26 | 65.15±20.95 |
| | (CV %) | 29.72 | 36.04 | 23.32 | 32.16 |

Note: SD – standard deviation; CV (%) – co-efficient of variation (%); LFS – Low fertility strip: MFS – Medium fertility strip: HFS – High fertility strip.

## Soil sample analysis tests

| Test Parameter | Method Used | Reference |
|---|---|---|
| Available Nitrogen | Alkaline KMnO$_4$ method | [20] |
| Available Phosphorus | Bray's extractant (0.025 M HCl+0.03 M NH$_4$F), colorimetry by ascorbic acid method | [21] |
| Available Potassium | 1 N ammonium acetate extraction, Flame photometry | [22] |

## Plant sample analysis tests

| Parameter | Method Used | Reference (Standard Protocol) |
|---|---|---|
| Nitrogen Content | Micro Kjeldahl method | [23] |
| Phosphorus Content | Vanadomolybdate phosphoric acid yellow color method | [16] |
| Potassium Content | Flame photometry | [16] |
| Total Phenols | Folin-Ciocalteu colorimetric method | [24] |
| Total Flavonoids | Aluminum chloride colorimetric assay | [25] |
| Ascorbic Acid | AOAC titration/spectrophotometric method | [26] |

## Quality parameters of coriander

**Nutrient use efficiency indices.** To assess the nutrient use efficiency (NUE) of fertilizers, various parameters were analyzed, including partial factor productivity (PFP), agronomic efficiency (AE), partial nutrient balance (PNB), and internal utilization efficiency (IUE) [27].

Partial factor productivity (kg kg$^{-1}$) is determined by calculating the ratio of crop yield (kg) to the quantity of nutrient applied (kg).

$$PFP = \frac{\left[ \text{Yield in fertilized plot} \left( \text{kg ha}^{-1} \right) \right]}{\text{Nutrient applied} \left( \text{kg ha}^{-1} \right)}$$

Agronomic efficiency (kg kg$^{-1}$) is calculated as the increase in green foliage yield (kg) compared to the control, per unit of nutrient applied (kg).

$$AE = \frac{\left[\text{Yield in fertilized plot} - \text{Yield in control plot} \left(\text{kg ha}^{-1}\right)\right]}{\text{Nutrient applied (kg ha}^{-1})}$$

Partial nutrient budget (kg kg$^{-1}$) serves as an indicator for assessing the long-term sustainability of a cropping system, based on the nutrient uptake by plants per unit of nutrient applied.

$$PNB = \frac{\left[\text{Nutrient uptake in fertilized plot} \left(\text{kg ha}^{-1}\right)\right]}{\text{Nutrient applied (kg ha}^{-1})}$$

Internal utilization efficiency (IUE) represents the green foliage yield generated per kilogram of nutrient accumulated in the above-ground dry matter of the plant. A high IUE suggests a deficiency of a specific nutrient, while a low IUE indicates the plant's inability to convert nutrients into economic yield due to various stress factors, including deficiencies of other nutrients (antagonistic effects), drought, heat stress, mineral toxicities, and pest infestations.

$$IUE = \frac{\left[\text{Yield in fertilized plot} \left(\text{kg ha}^{-1}\right)\right]}{\text{Nutrient uptake in fertilized plot (kg ha}^{-1})}$$

### Statistical analysis

The fundamental data obtained from the test-crop experiment served as the basis for formulating fertilizer prescription equations. The established fertility gradient was visually represented using a box and whiskers plot, and the data from the test-crop experiment were summarized using descriptive statistics with SPSS 16.0 software. The validation trial data were subjected to analysis of variance (ANOVA) using a randomized block design [28], with statistical computations carried out via the OPSTAT online program. Treatment means were separated and compared using Tukey's Least Significant Difference (LSD) test at the 5% significance level. Pearson correlation coefficient (r) values were calculated using Microsoft Excel 2016. Microsoft Excel (MS Office 10.0) and Origin plot software (version 2020b 9.75) were utilized for the creation of figures and graphs presented in the manuscript.

## Results

### Fertility gradient experiment with fodder maize

Post-harvest soil nutrient values derived from the fertility strips, as shown in Fig 2, were illustrated using a box-and-whisker plot to depict the variability and distribution across different fertility levels. The post-harvest soil test values for NPK increased with an increase in applied nutrient levels from low fertility strip (LFS- 245.56, 149.51 and 230.54 kg N, P$_2$O$_5$, and K$_2$O ha$^{-1}$) to high fertility strip (HFS- 299.71, 181.37 and 271.93 kg N, P$_2$O$_5$ and K$_2$O ha$^{-1}$), which indicates the development of fertility gradient. A detailed analysis of post-harvest soil-available nutrients—nitrogen, phosphorus, and potassium reveals intriguing trends in their distribution and skewness across LFS, MFS, and HFS (Fig 2). Nitrogen displays relative symmetry in LFS and MFS, with slight positive skewness in HFS, suggesting occasional higher concentrations. Phosphorus, on the other hand, shows gradual positive skewness as levels progress, with HFS exhibiting a pronounced skew due to higher values, potentially influenced by specific experimental factors. Potassium exhibits stable and nearly symmetrical distributions in the LFS and MFS, but is markedly positively skewed in the HFS, reflecting sporadically high concentrations due to external influences. These influences include elevated potassic fertilizer inputs, improved nutrient retention, suitable moisture and temperature, heightened microbial activity from organic amendments,

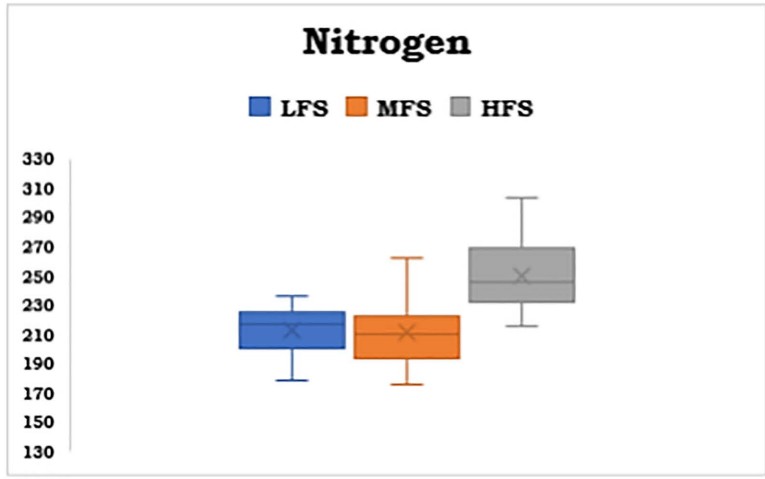

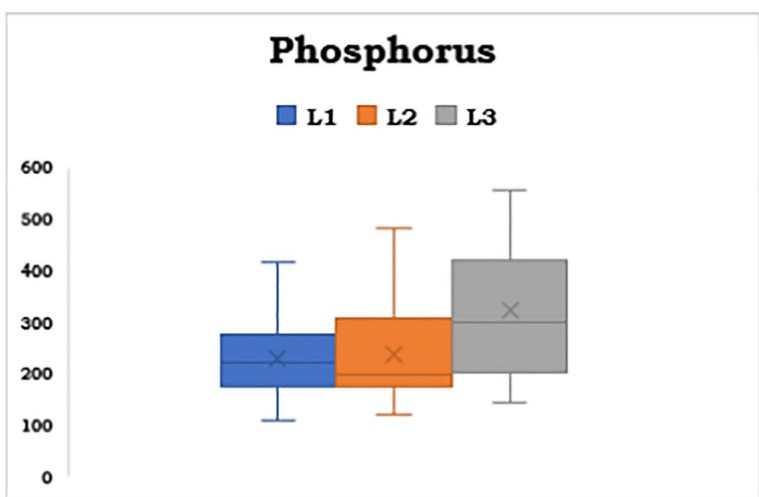

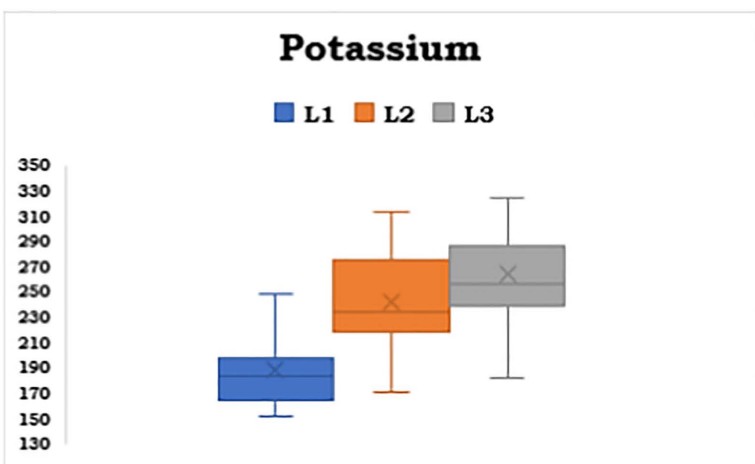

**Fig 2. Box and whisker plots of the available soil nutrients after the soil fertility gradient experiment.**

and increased crop root activity that mobilizes more soil potassium. Collectively, these factors result in variable potassium availability in managed sandy loam soils, as confirmed by published research.These patterns of positive skewness across HFS indicates the role of environmental or experimental factors in elevating nutrient levels, emphasizing the importance of addressing variability for optimized nutrient management.

**Main experiment with coriander**

The data in Table 2 and Fig 3 summarize yield and nutrient uptake across fertility strips (LFS, MFS, HFS) in the coriander main experiment. Test-crop data were analyzed with descriptive statistics, highlighting trends in yield and nutrient uptake. The descriptive statistics highlight the significant influence of fertility levels on coriander's green foliage yield and nutrient uptake. High fertility strips demonstrate superior performance, with green foliage yield ranging from 3.36 to 11.36 t ha$^{-1}$ (mean: 7.57 t ha$^{-1}$), nitrogen uptake from 21.13 to 71.18 kg ha$^{-1}$ (mean: 49.22 kg ha$^{-1}$), phosphorus uptake from 10.09 to 30.40 kg ha$^{-1}$ (mean: 22.56 kg ha$^{-1}$), and potassium uptake from 29.17 to 104.62 kg ha$^{-1}$ (mean: 65.15 kg ha$^{-1}$). In comparison, low fertility strips exhibit lower values, such as phosphorus uptake ranging from 5.24 to 23.32 kg ha$^{-1}$ (mean: 15.43 kg ha$^{-1}$). The coefficient of variation (CV%) provides insights into the consistency of yield and nutrient uptake across different fertility strips. Among the strips, the High Fertility Strip consistently exhibits lower CV values for most parameters, indicating greater stability and uniformity in performance. Fig 3 illustrates the relationship between yield and nutrient uptake in coriander, revealing a strong and nearly linear association. The data shows a significant correlation ($p < 0.001$), with total potassium uptake displaying the highest correlation ($r^2 = 0.96$), followed by total nitrogen uptake ($r^2 = 0.89$) and total phosphorus uptake ($r^2 = 0.71$). This emphasizes the interconnectedness of nutrient absorption and crop productivity.

The results from Table 3 demonstrate the differences in basic parameters for nutrient management in coriander under two scenarios: NPK alone and NPK combined with FYM (farmyard manure). The nutrient requirement and nutrient contribution from fertilizer values for nitrogen, phosphorus, and potassium are higher when FYM was included along with NPK fertilizers compared to the sole NPK application. The nutrient requirement for nitrogen increased by approximately 7.38%, from 6.10 kg q$^{-1}$ to 6.55 kg q$^{-1}$, while phosphorus showed a rise of 13.90%, from 2.23 kg q$^{-1}$ to 2.54 kg q$^{-1}$. Potassium exhibited the highest increase in NR, with a rise of 12.50%, from 7.91 kg q$^{-1}$ to 8.90 kg q$^{-1}$. In terms of nutrient contribution from fertilizers, nitrogen saw an increase of approximately 34.06%, phosphorus rose by 34.35%, and potassium demonstrated a substantial rise of 42.50%, from 91.45% to 130.35%. The contribution of nutrients from the soil is expressed as the capacity of the crop to extract nutrients from the soil. The percent contributions of NPK from the soil were 10.67, 11.66 and 10.40, respectively. Additionally, the inclusion of FYM introduced contributions from organic manure. The contributions of NPK from the FYM were 0.075, 0.095 and 0.072%, respectively. These results underscore the significant enhancements in nutrient availability and uptake efficiency when FYM is integrated with chemical fertilizers.

Based on the basic parameters such as nutrient requirements, fertilizer efficiency, soil test results, and contributions from organic sources, fertilizer adjustment equations were precisely developed to achieve targeted yield goals in coriander. These equations were calibrated to optimize nutrient management practices, ensuring efficient resource utilization and improved productivity for the crop.

**STCR-NPK equations alone.**

$$FN = 18.31T - 0.32\ SN \tag{1}$$

$$FP = 8.30\ T - 0.43\ SP \tag{2}$$

$$FK = 8.64\ T - 0.11\ SK \tag{3}$$

**STCR-NPK + FYM equations.**  $$FN = 14.66\ T - 0.24\ SN - 0.08\ OM \tag{4}$$

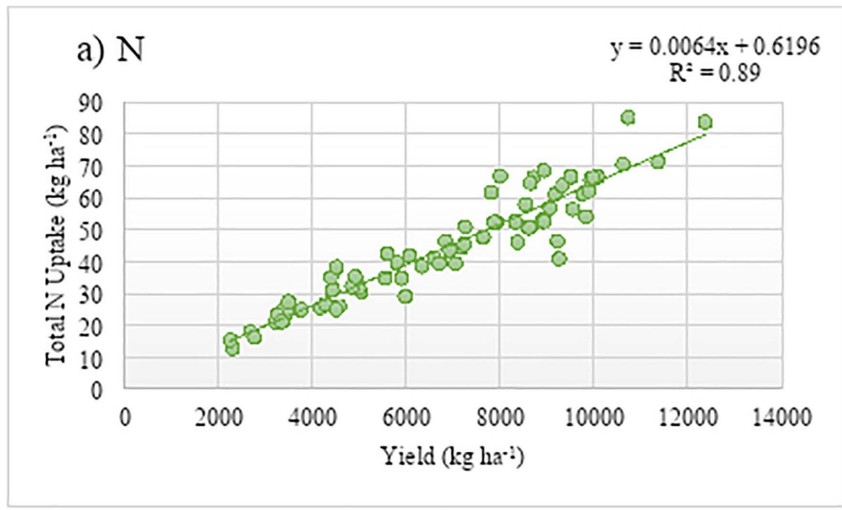

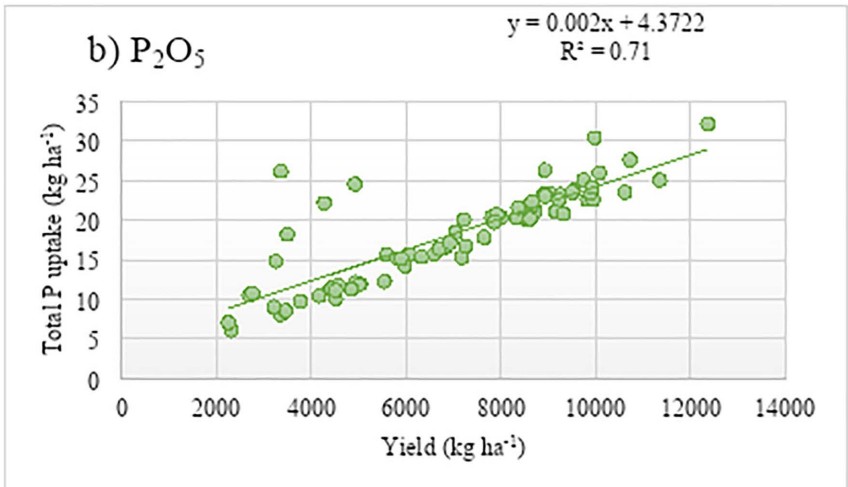

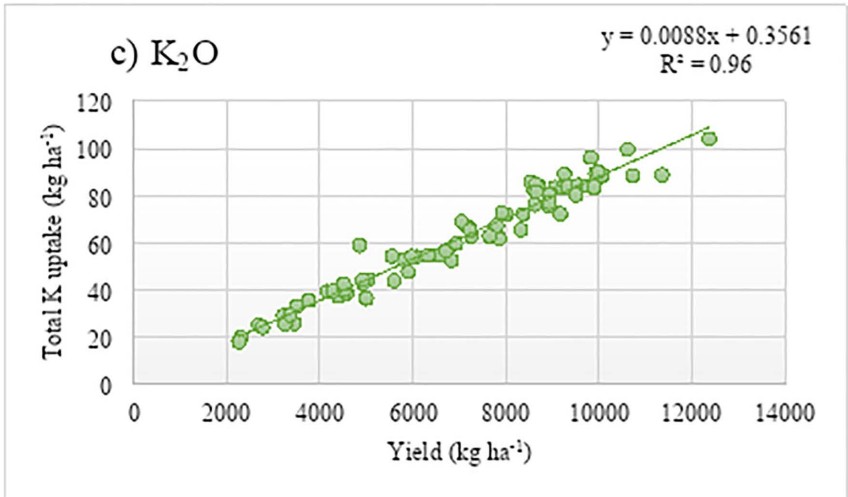

**Fig 3. Linear correlations between green foliage yield and total N (a), P2O5 (b), and K2O (c) uptake in coriander in the main experiment.**

**Table 3. Basic parameters for equation development.**

| Parameters | NPK alone | | | NPK+FYM | | |
|---|---|---|---|---|---|---|
| | N | $P_2O_5$ | $K_2O$ | N | $P_2O_5$ | $K_2O$ |
| NR (kg q$^{-1}$) | 6.10 | 2.23 | 7.91 | 6.55 | 2.54 | 8.90 |
| CS% | 10.67 | 11.66 | 10.40 | 10.67 | 11.66 | 10.40 |
| CF% | 33.32 | 26.90 | 91.45 | 44.68 | 36.14 | 130.35 |
| COM% | – | – | – | 0.075 | 0.095 | 0.072 |

Note: NR-Nutrient requirement: CS-Contribution of nutrients from soil: CF-Contribution of nutrients from fertilizers: COM-Contribution of nutrients from organic manure.

$$FP = 7.03\ T - 0.32\ SP - 0.09\ OM \tag{5}$$

$$FK = 6.83\ T - 0.08\ SK - 0.07\ OM \tag{6}$$

where FN, FP and FK are the N, P and K fertilizers in kg ha$^{-1}$, respectively; T is the yield target at t ha$^{-1}$; SN, SP and SK are the available soil nutrients in kg ha–1; and OM is the amount of organic manure (FYM) added at t ha$^{-1}$.

The regression relationships in Table 4 illustrate the multivariate model linking crop yield to soil available nutrients (N, $P_2O_5$, and $K_2O$). The derived regression equation indicates that yield is positively influenced by all three nutrients, with the coefficients showing their respective contributions: nitrogen (19.53), phosphorus (23.89), and potassium (9.65). The model explains 50.6% of the variation in yield ($r^2 = 0.506$), and the adjusted $R^2$ value (0.480) confirms the model's reliability after accounting for the number of predictors. The regression is highly significant ($p < 0.001$), with individual nutrients showing varied levels of significance. Nitrogen has the strongest effect, as indicated by its t-statistic (4.20) and a highly significant p-value (< 0.001). Phosphorus and potassium also contribute significantly, with t-statistics of 3.10 and 2.15, and p-values $< 0.05$. These findings emphasize the critical roles of soil nitrogen, phosphorus, and potassium in influencing crop yield and validate the strength of the regression model in explaining their combined impact.

**Table 4. Multivariate regression models to explain the association between the yield (Y) and soil available nutrients (N, P and K) under main experiment.**

| Regression equation | | $Y^* = -3891.25 + 19.53\ SN + 23.89\ SP + 9.65\ SK$ |
|---|---|---|
| Observations (*n*) | | 72 |
| $R^2$ | | 0.506 |
| Adjusted $R^2$ | | 0.480 |
| Regression (*p* value) | | <0.001 |
| *t*–statistics | N | 4.20 |
| | $P_2O_5$ | 3.10 |
| | $K_2O$ | 2.15 |
| *p* value | N | <0.001 |
| | $P_2O_5$ | <0.05 |
| | $K_2O$ | <0.05 |

*Y – Yield: SN – Soil available N: SP – Soil available $P_2O_5$: SK – Soil available $K_2O$.

From the ready reckoner (Table 5) it was observed that to achieve the yield target of 8 t ha$^{-1}$ of coriander with the soil test values of 280:23:140 kg ha$^{-1}$ for available N: $P_2O_5$:$K_2O$, the recommended dose of fertilizer nutrients to be applied are 56.90:56.45:53.25 kg ha$^{-1}$ without FYM and 49.90:48.04:42.89 kg ha$^{-1}$ along with 7.5 t ha$^{-1}$ of FYM. Similarly, when the soil test values are high, i.e., 560:52:320 kg ha$^{-1}$ of N: $P_2O_5$:$K_2O$, the fertilizer nutrients required for the same yield target is −32.80:43.88:35.05 kg ha$^{-1}$ without FYM and −17.00:38.71:30.13 kg ha$^{-1}$ with 7.5 t ha$^{-1}$ of FYM. The negative values indicate that the application of respective fertilizers is not required, but 25–30 percent of the recommended doses of fertilizers are recommended to maintain the soil fertility status.

### Validation experiment

**Green foliage yield and nutrient uptake.** Table 6 summarizes the mean values for yield, quality, and nutrient uptake across three replications. Statistical analysis using ANOVA, followed by Tukey's LSD test at the 5% significance level, revealed that STCR-based treatments ($T_1$–$T_4$) consistently outperformed the General Recommended Dose (GRD) and Soil Fertility Rating (SFR) methods in enhancing coriander yield and nutrient uptake.). The highest green foliage yield of coriander was recorded in STCR TY 10 t ha$^{-1}$ with NPK + FYM (11.15 t ha$^{-1}$), which recorded a 54.23% and 56.90% higher yield than GRD and SFR, respectively. Additionally, STCR approach with NPK + FYM for the targeted yield of 10 t ha$^{-1}$ exhibited the highest nutrient uptake across all parameters, with nitrogen uptake increasing by 32.25% over GRD and 48.91% over SFR, phosphorus uptake by 40.71% over GRD and 33.55% over SFR, and potassium uptake by 70.30% over GRD and 85.78% over SFR. Similarly, STCR TY 10 t ha$^{-1}$ through NPK alone also showed a significant increase in yield and nutrient uptake, with yield rising 44.14% over GRD and 46.76% over SFR, and notable improvements in nitrogen,

**Table 5. Ready Reckoner indicating fertilizer recommendation for coriander with a targeted yield of 8 t ha$^{-1}$.**

| Soil test values | | | NPK alone | | | NPK + 7.5 FYM ha$^{-1}$ | | |
|---|---|---|---|---|---|---|---|---|
| N | $P_2O_5$ | $K_2O$ | F. N. | F. $P_2O_5$ | F. $K_2O$. | F. N. | F.$P_2O_5$ | F. $K_2O$. |
| ---------------------- kg ha$^{-1}$ ------------------------- | | | | | | | | |
| 250 | 15 | 110 | 66.5 | 59.92 | 56.67 | 57.0 | 50.65 | 45.29 |
| 260 | 17 | 120 | 63.3 | 59.05 | 55.53 | 54.6 | 50.01 | 44.49 |
| 280 | 19 | 125 | 56.9 | 58.18 | 54.96 | 49.9 | 49.36 | 44.09 |
| 290 | 21 | 130 | 53.7 | 57.32 | 54.39 | 47.5 | 48.72 | 43.69 |
| 300 | 23 | 140 | 50.4 | 56.45 | 53.25 | 45.1 | 48.07 | 42.89 |
| 320 | 25 | 150 | 44.0 | 55.58 | 52.12 | 40.3 | 47.43 | 42.10 |
| 340 | 30 | 160 | 37.6 | 53.41 | 50.98 | 35.5 | 45.81 | 41.30 |
| 400 | 32 | 170 | 18.4 | 52.55 | 49.84 | 21.2 | 45.17 | 40.50 |
| 440 | 35 | 180 | 5.6 | 51.25 | 48.70 | 11.6 | 44.20 | 39.70 |
| 480 | 38 | 200 | −7.2 | 49.95 | 46.43 | 2.1 | 43.23 | 38.11 |
| 520 | 41 | 220 | −20.0 | 48.65 | 44.15 | −7.5 | 42.26 | 36.51 |
| 560 | 44 | 240 | −32.8 | 47.34 | 41.88 | −17.0 | 41.29 | 34.91 |
| 600 | 46 | 260 | −45.6 | 46.48 | 39.61 | −26.6 | 40.65 | 33.32 |
| | 47 | 280 | | 46.04 | 37.33 | | 40.33 | 31.72 |
| | 50 | 300 | | 44.74 | 35.06 | | 39.36 | 30.13 |
| | 52 | 320 | | 43.88 | 32.78 | | 38.71 | 28.53 |
| | 54 | 340 | | 43.01 | 30.51 | | 38.07 | 26.93 |
| | 58 | 440 | | 41.27 | 19.13 | | 36.78 | 18.96 |

F.N.: Fertilizer nitrogen, F.$P_2O_5$: Fertilizer phosphorus, F.$K_2O$: Fertilizer potassium.

**Table 6. Influence of different approaches of fertilizer recommendation on yield, nutrient uptake and quality of coriander.**

| Treatment | Green foliage yield (t ha⁻¹) | Total phenols (mg/ g sample) | Total flavonoids (mg/ g sample) | Ascorbic acid (mg/100g) | Nutrient uptake (kg ha⁻¹) | | |
|---|---|---|---|---|---|---|---|
| | | | | | N | P₂O₅ | K₂O |
| $T_1$ | 10.42$^a$ | 2.61$^a$ | 15.09$^b$ | 210.58$^c$ | 61.75$^b$ | 18.53$^a$ | 62.63$^b$ |
| $T_2$ | 11.15$^b$ | 2.92$^b$ | 16.12$^a$ | 220.14$^a$ | 65.95$^a$ | 18.10$^c$ | 74.67$^a$ |
| $T_3$ | 8.19$^c$ | 2.30$^c$ | 12.67$^d$ | 198.46$^d$ | 50.69$^d$ | 14.96$^d$ | 53.83$^d$ |
| $T_4$ | 8.74$^c$ | 2.52$^d$ | 13.49$^c$ | 215.84$^b$ | 57.45$^c$ | 18.19$^b$ | 46.21$^c$ |
| $T_5$ | 7.23$^d$ | 1.96$^e$ | 12.12$^e$ | 108.69$^e$ | 49.87$^e$ | 12.86$^e$ | 43.84$^e$ |
| $T_6$ | 7.10$^d$ | 1.81$^f$ | 11.98$^f$ | 98.47$^f$ | 44.59$^f$ | 13.56$^f$ | 40.21$^f$ |
| $T_7$ | 3.51$^e$ | 1.02$^g$ | 7.60$^g$ | 32.10$^g$ | 23.06$^g$ | 6.62$^g$ | 22.70$^g$ |
| SEm+ | 0.67 | 0.09 | 0.51 | 17.10 | 2.01 | 0.99 | 5.02 |
| CD @ (5%) | 1.91 | 0.23 | 1.28 | 42.75 | 5.75 | 2.81 | 14.29 |
| CV (%) | 5.32 | 4.28 | 4.42 | 6.24 | 5.14 | 6.48 | 6.27 |

Note: $T_1$- STCR TY 10 t ha⁻¹ (NPK Alone); $T_2$- STCR TY 10 t ha⁻¹ (NPK+FYM); $T_3$- STCR TY 8 t ha⁻¹ (NPK Alone); $T_4$- STCR TY 8 t ha⁻¹ (NPK+FYM); $T_5$- General recommended dose; $T_6$- Soil fertility rating; $T_7$- Absolute control.

a,b,c,d,e,f,gTreatment means followed by different letters are significantly different from each other at the 5% level according to Tukey's Least Significant Difference (LSD) test.

phosphorus, and potassium uptake. Notably, absolute control, (without fertilizers) produced the lowest yield (3.51 t ha⁻¹) and exhibited minimal nutrient uptake, underscoring the necessity of fertilizer application in coriander production.

**Quality parameters of coriander.** The study highlights the significant impact of different fertilizer recommendation approaches on coriander's quality parameters, including total phenols, flavonoids, and ascorbic acid content. Among all treatments, STCR TY 10 t ha⁻¹ with NPK+FYM exhibited the highest quality values, with total phenols increasing by 48.98% over GRD and 61.33% over SFR, flavonoids increasing by 32.99% over GRD and 34.18% over SFR, and ascorbic acid improving by 102.53% over GRD and 123.53% over SFR, demonstrating that organic supplementation (FYM) enhances biochemical attributes. Similarly, STCR TY 10 t ha⁻¹ with NPK alone showed substantial improvement, confirming the effectiveness of precise fertilizer management. In contrast, the general recommended dose and soil fertility rating approach recorded lower values, indicating that conventional fertilization does not maximize secondary metabolite production. The lowest biochemical values were observed in absolute control, highlighting the crucial impact of fertilization on enhancing the nutritional quality of coriander.

**Percent deviation, response yard stick (RYS) and value cost ratio (VCR).** The yield obtained from STCR treatments targeting 10 t ha⁻¹ and 8 t ha⁻¹ with NPK alone (showing positive deviations of 4.20% and 11.50%, respectively) and NPK+FYM (2.37% and 9.25%, respectively) surpassed the fixed yield targets, as illustrated in Fig 4. The efficiency of nutrient use, measured as response yield, varied considerably among treatments, ranging from 35.43 to 53.15 kg kg⁻¹ (Fig 4). In contrast to the STCR methods, the soil fertility rating approach (37.30 kg kg⁻¹) and general fertilizer recommendations (35.43 kg kg⁻¹) exhibited lower RYS values. The most economically beneficial treatment, with a value cost ratio of 81.87, was STCR NPK for an 8 t ha⁻¹ target, closely followed by the 10 t ha⁻¹ STCR NPK treatment (VCR of 80.97) (Fig 4). The soil fertility rating approach showed the least favorable VCR (10.79). Furthermore, the integration of FYM into STCR treatments resulted in a lower VCR compared to using only NPK.

**Nutrient use efficiency of coriander.** The study highlights the influence of different fertilizer recommendation approaches on nutrient use efficiency in coriander, considering Partial Factor Productivity, Agronomic Efficiency, Partial Nutrient Budgeting, and Internal Utilization Efficiency (Table 7). Among all treatments, STCR TY 8 t ha⁻¹ with NPK+FYM recorded the highest PFP for phosphorus (610.05 kg kg⁻¹), AE for phosphorus (365.05 kg kg⁻¹), and PNB for phosphorus (1.27 kg kg⁻¹), demonstrating that lower target yields with organic supplementation enhance nutrient efficiency, particularly phosphorus utilization. Meanwhile,

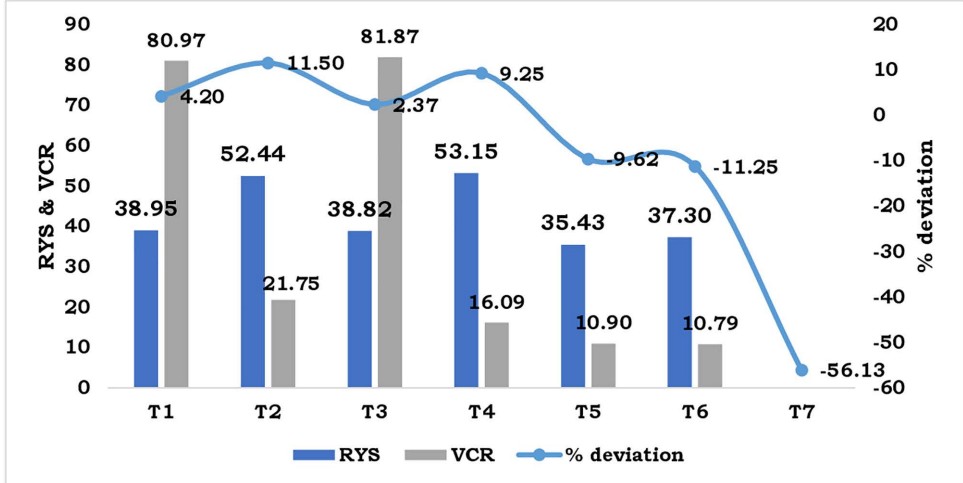

**Fig 4. Influence of different approaches of fertilizer recommendation on RYS, VCR and % deviation of coriander.**

**Table 7. Influence of different approaches of fertilizer recommendation on nutrient use efficiency of coriander.**

| Treatments | PFP (kg kg⁻¹) | | | AE (kg kg⁻¹) | | | PNB (kg kg⁻¹) | | | IUE | | |
|---|---|---|---|---|---|---|---|---|---|---|---|---|
| | N | $P_2O_5$ | $K_2O$ | N | $P_2O_5$ | $K_2O$ | N | $P_2O_5$ | $K_2O$ | N | $P_2O_5$ | $K_2O$ |
| $T_1$ | 123.64 | 305.60 | 177.44 | 81.99 | 202.66 | 117.67 | 0.73 | 0.54 | 1.06 | 169.03 | 563.27 | 166.65 |
| $T_2$ | 163.82 | 372.37 | 235.04 | 112.25 | 255.15 | 161.05 | 0.97 | 0.60 | 1.57 | 169.35 | 617.05 | 149.57 |
| $T_3$ | 143.38 | 391.03 | 193.66 | 81.93 | 223.45 | 110.66 | 0.89 | 0.71 | 1.27 | 161.84 | 548.37 | 152.40 |
| $T_4$ | 181.66 | 610.05 | 244.18 | 108.71 | 365.05 | 146.12 | 1.19 | 1.27 | 1.29 | 152.39 | 481.28 | 189.45 |
| $T_5$ | 206.92 | 206.92 | 206.92 | 106.46 | 106.46 | 106.46 | 1.42 | 0.37 | 1.25 | 145.22 | 563.15 | 165.19 |
| $T_6$ | 203.20 | 270.93 | 203.20 | 102.74 | 136.99 | 102.74 | 1.27 | 0.52 | 1.15 | 159.49 | 524.47 | 176.87 |
| $T_7$ | – | – | – | – | – | – | – | – | – | 152.47 | 531.10 | 154.88 |

Note: PFP: Partial factor productivity, AE: Agronomic efficiency, PNB: Partial Nutrient budgeting, IUE: Internal utilization efficiency.

$T_1$- STCR TY 10 t ha⁻¹ (NPK Alone); $T_2$- STCR TY 10 t ha⁻¹ (NPK+FYM); $T_3$- STCR TY 8 t ha⁻¹ (NPK Alone); $T_4$- STCR TY 8 t ha⁻¹ (NPK+FYM); $T_5$- General recommended dose; $T_6$- Soil fertility rating; $T_7$- Absolute control.

STCR TY 10 t ha⁻¹ with NPK+FYM showed the highest nitrogen PFP (163.82 kg kg⁻¹) and potassium PFP (235.04 kg kg⁻¹), along with improved agronomic efficiency values, indicating that higher target yields with integrated nutrient management enhance productivity per unit of fertilizer applied. The GRD and SFR approaches showed lower PNB values for phosphorus (0.37 and 0.52 kg kg⁻¹, respectively) and moderate agronomic efficiency compared to STCR-based recommendations, reinforcing the importance of site-specific nutrient management for optimal fertilizer utilization.

Notably, STCR TY 10 t ha⁻¹ with NPK alone and STCR TY 8 t ha⁻¹ with NPK alone showed efficiency improvements but performed slightly lower than their counterparts supplemented with FYM, emphasizing the beneficial role of organic inputs in nutrient availability. The highest IUE of nitrogen was recorded with STCR TY 10 t ha⁻¹ with NPK+FYM, while the potassium utilization efficiency was highest in STCR TY 8 t ha⁻¹ with NPK+FYM, demonstrating the role of FYM in optimizing nutrient internal utilization within plants. The absolute control recorded the lowest nutrient efficiency across all parameters, underscoring the essential role of fertilization in improving nutrient use efficiency and overall crop sustainability.

Fig 5, 6, 7 show significant Pearson correlations (r, $p < 0.05$) between soil organic carbon (SOC) and nutrient use efficiency indices, calculated from replicated validation experiment data. Linear regression analysis across fertilizer strategies

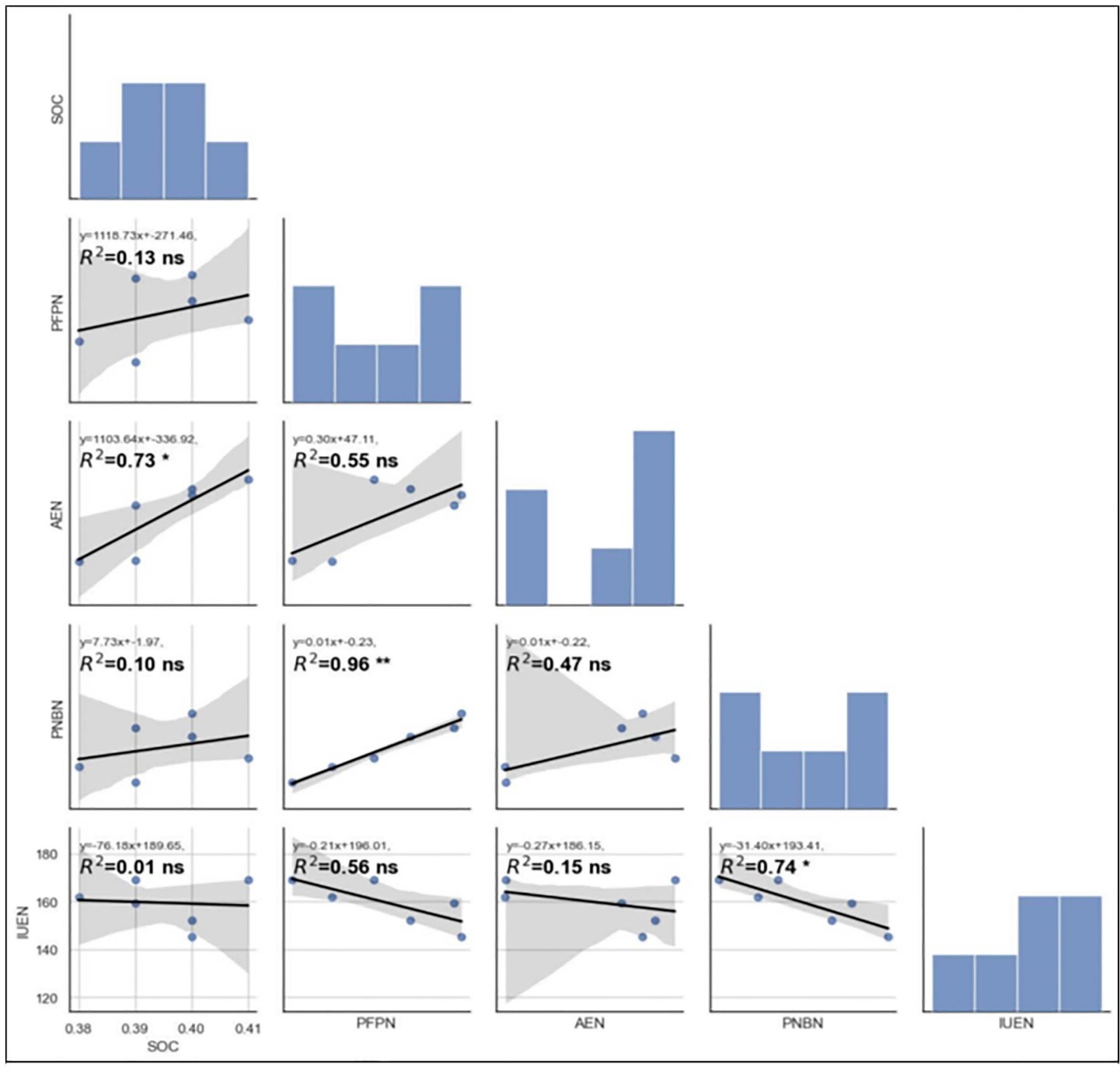

**Fig 5. Correlation of soil organic carbon with different nutrient use efficiencies of nitrogen as affected by different approaches of fertilizer prescription.** SOC- soil organic carbon; AE- Agronomic efficiency; PFP-partial factor productivity; PNB-partial nutrient balance; IUE-internal utilization efficiency.

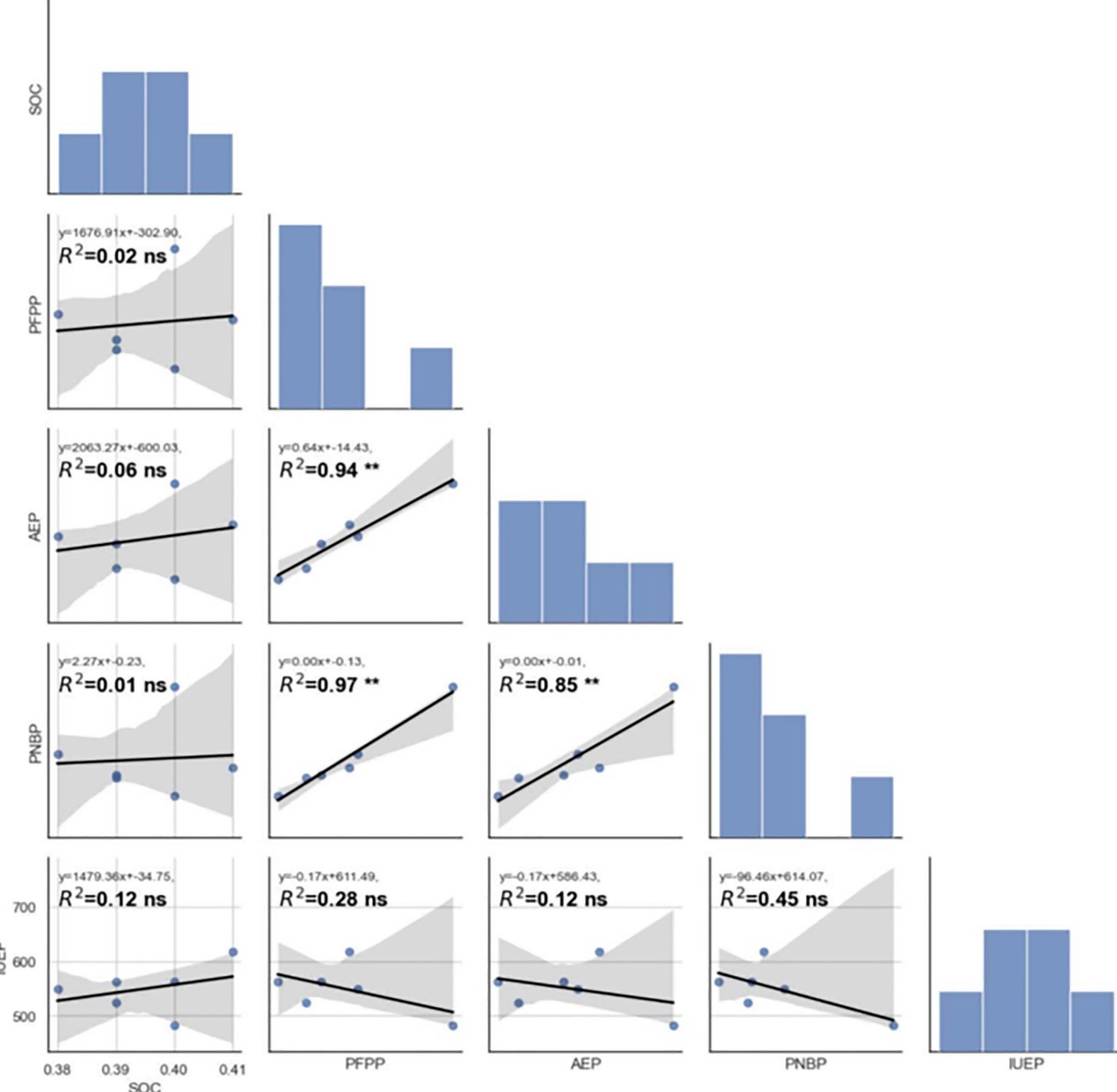

**Fig 6. Correlation of soil organic carbon with different nutrient use efficiencies of phosphorus as affected by different approaches of fertilizer prescription. SOC- soil organic carbon; AE- Agronomic efficiency; PFP-partial factor productivity; PNB-partial nutrient balance; IUE-internal utilization efficiency.**

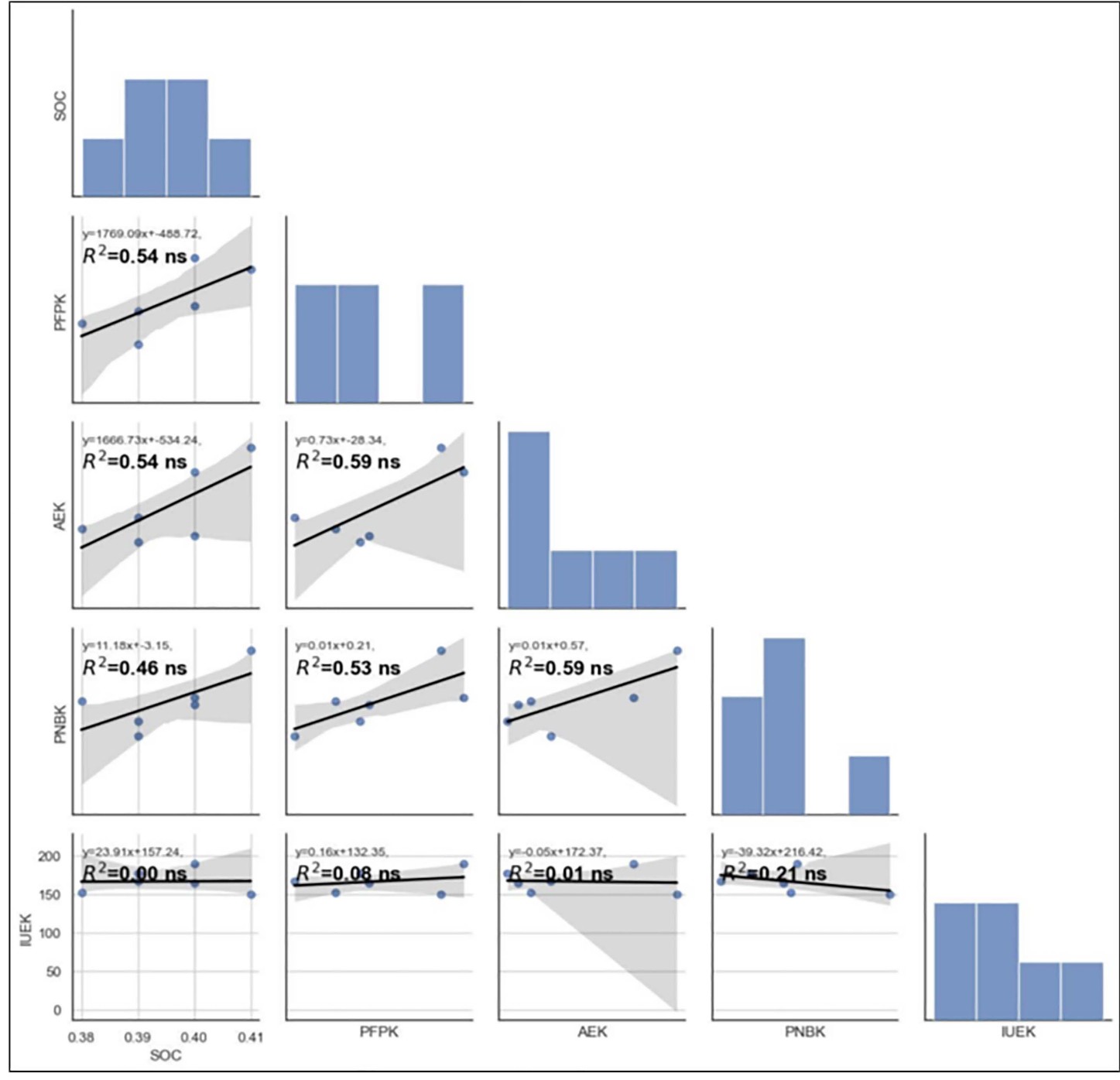

**Fig 7. Correlation of soil organic carbon with different nutrient use efficiencies of potassium as affected by different approaches of fertilizer prescription.** SOC- soil organic carbon; AE- Agronomic efficiency; PFP-partial factor productivity; PNB-partial nutrient balance; IUE-internal utilization efficiency.

(STCR, GRD, SFR) revealed strong positive relationships between SOC and agronomic efficiency (AE), partial factor productivity (PFP), and partial nutrient balance (PNB) of nitrogen and phosphorus under STCR treatments. Coefficients of determination ($R^2$) ranged from 0.73* to 0.96**, indicating robust predictive associations ($p < 0.05$ and $p < 0.01$).Conversely, SOC exhibited only low or non-significant correlations with IUE for NPK and PNB for potassium when organic manures were included, as denoted by low $R^2$ values ($\leq$ 0.13 ns), revealing minimal explanatory power and statistical insignificance ($p \geq 0.05$). Histograms and pairwise scatterplots further emphasized these findings, with integrated STCR management outperforming other approaches in enhancing nutrient use efficiency and crop productivity, as indicated by the best fit lines and statistical significance (Fig 5, 6, 7).

## Discussion

This study presents the first STCR calibration equations tailored for coriander cultivation on southern Indian Alfisols, uniquely incorporating farmyard manure coefficients within fertilizer calibration and validating them through a ready-reckoner in field trials. These advances enable precise nutrient management specific to coriander and local conditions, unlike prior general approaches. However, since the nutrient coefficients and equations were developed for southern Karnataka's Alfisols and climate, their applicability to other regions or soils is limited without adaptation and recalibration. Future research should focus on multi-location trials across diverse soils and climates to refine these models, facilitating wider adoption of precise and sustainable nutrient management for coriander and potentially other crops.

The fertility gradient experiment demonstrated that incremental nutrient inputs effectively increased post-harvest availability of N, P, and K, as reflected by soil test values across low, medium, and high fertility strips. Rather than merely restating results, these findings show that higher fertility levels are achieved through targeted nutrient management, confirming the soil's clear responsiveness and capacity to accumulate nutrients when application rates are matched to crop demand and soil characteristics [29]. However, the positive skewness and variability found at higher fertility levels indicate that not all plots respond uniformly, underscoring the challenges of ensuring consistent nutrient distribution in practice. These patterns of nutrient variability highlight the influence of environmental heterogeneity and management practices, and reveal that soil nutrient reservoirs, especially for nitrogen, phosphorus, and potassium, can exhibit significant differences even under the same input regimes [7]. Enhanced available nitrogen in high fertility strips may be attributed to improved $NH_4^+$ adsorption on organic and inorganic colloids [30]. Elevated phosphorus levels likely occur when fertilizer additions surpass the soil's P fixation capacity, increasing P in solution [31]. Enhanced potassium may result from applications that exceed the soil's K fixation threshold, allowing for retention in the exchangeable form [32].

The descriptive statistics reveal that fertility levels significantly impact the yield and nutrient uptake of coriander. These variations are crucial for establishing soil fertility gradients and are necessary for computing basic parameters and formulating fertilizer recommendation equations. High fertility strips consistently show superior performance in green foliage yield, nitrogen, phosphorus, and potassium uptake compared to low and medium fertility strips. Moreover, the variability in these metrics tends to decrease with higher fertility, indicating more consistent and stable results. This underscores the critical role of fertility management in optimizing yield and nutrient utilization in coriander cultivation [33]. The reduced variation (CV%) in HFS highlights its efficacy in providing consistent yields and nutrient uptake, underscoring the value of improved fertility management in coriander cultivation. The combined application of NPK fertilizers and farmyard manure (FYM) further enhanced coriander productivity compared to NPK alone. This can be attributed not only to the direct supply of nutrients through FYM decomposition and mineralization, but also to the indirect stimulation of rhizosphere microbial activity, which improves nutrient availability [3]. Such integration promotes root proliferation, improving the plant's capacity to absorb both water and nutrients, an effect well-documented under NPK plus FYM management regimes [19,34].Studies have shown that integrating the STCR approach with both inorganic fertilizers and farmyard manure markedly improves coriander yield and nutrient uptake efficiency [35,36].

The targeted yield model developed in this study, calibrated through soil testing, highlights the merit of integrating pre-sowing soil nutrient levels, total nutrient uptake, and the combined application of N, P, K fertilizers along with farmyard manure for coriander. The nutrient requirements derived from these models reveal a descending hierarchy of $K_2O > N > P_2O_5$, aligning with previous observations in [37] and [38]. This underscores the critical need for potassium management in coriander cultivation.When examining the three major nutrients, the soil's inherent supply was most significant for nitrogen, followed by phosphorus, and then potassium. However, the trend in nutrient contribution from the soil itself was K>N>P, potentially due to the preferential uptake of nutrients by the crops [39]. Conversely, the order of percentage contribution of fertilizer nutrients to crop uptake and yield under both inorganic and integrated nutrient management was K>P>N. Fertilizers contributed a larger proportion of nutrients than the soil, likely because inorganic fertilizers provide a more readily available and concentrated source of nutrients [40]. Interestingly, the integrated approach showed a greater contribution of nutrients from fertilizers compared to using only inorganic fertilizers. Notably, potassium contribution from fertilizers sometimes exceeded 100%, an anomaly attributed to the combined effect of higher nitrogen and phosphorus application at a specific K level, alongside the direct impact of initial K application, which appeared to trigger the release of soil-bound K, leading to increased uptake from native soil reserves in the presence of applied K [41]. The contribution from organic carbon, estimated using data from FYM-treated and control plots, followed the order N>P>K [42]. The variability and interactive effects observed when summing the percentages of nutrient contributions from soil, fertilizer, and organic sources [43] reflect complex nutrient cycling dynamics, emphasizing the need for site- and crop-specific calibration of input models for efficient nutrient management in coriander systems.The addition of 7.5 t FYM ha$^{-1}$ consistently reduces the fertilizer requirements for nitrogen, phosphorus, and potassium, as FYM acts as a supplementary source of nutrients. This demonstrates the role of FYM in improving nutrient efficiency and reducing dependency on synthetic fertilizers. Higher soil test values for NPK significantly decrease the external fertilizer input needed to achieve the targeted yield, highlighting the importance of soil fertility in nutrient management planning [37]. Integrating farmyard manure with inorganic fertilizer application led to a notable decrease in the required amounts of chemical fertilizer compared to using inorganic fertilizer alone. This reduction was attributable to the additional nutrients supplied by the FYM. Furthermore, the percentage decrease in fertilizer application increased with higher soil test values when FYM was used in conjunction with inorganic fertilizers [44]. For high soil test values, negative fertilizer requirements indicate surplus nutrient availability, where further fertilizer application may not only be unnecessary but could also risk nutrient losses or environmental harm [38].

Conducting a verification trial is crucial to assess the reliability of the calibration results obtained under main experiment before recommending the derived equations to farmers [45]. The results showed that crop nutrient management based on STCR approach integrated with the application of 7.5 t FYM ha$^{-1}$ significantly enhanced the yield and nutrient uptake of coriander. STCR-based approaches, particularly those incorporating organic amendments, significantly outperformed conventional fertilizer methods by optimizing yield, nutrient uptake, and resource efficiency [46]. This suggests that precision nutrient management strategies are key to sustainable and high-yielding coriander production. Studies by [47] and [48] confirm that higher NPK rates significantly enhance leaf yield, quality, and plant growth, while earlier research also links increased NPK with improved chlorophyll content and overall coriander productivity. Utilizing organic manure might have led to a notable improvement in the soil environment, encouraging root proliferation [49] and supporting enhanced plant growth. Moreover, the benefits of organic manures extend beyond nutrient provision to the creation of an environment that supports increased growth and activity of soil organisms, like the native rhizobial population, which might have improved crop yield [50]. Studies found that full recommended nitrogen doses, combined with vermicompost or FYM, markedly improve leaf growth and nitrogen uptake in coriander, particularly in Alfisols [51,52]. The reduced yield observed with the soil fertility rating approach and general recommended dose is likely due to the application of nutrients without proper consideration of the crop's specific requirements and the contributions from the soil, fertilizer, and FYM [5]. The enhanced uptake of nitrogen, phosphorus, and potassium in the STCR approach for the targeted yield of 10 t ha$^{-1}$ through

NPK + FYM can be attributed to the higher fertilizer dose, which likely contributed to better nutrient availability in the root zone, thereby improving uptake efficiency [53]. Higher applications of chemical NPK combined with biofertilizers consistently improved vegetative growth and significantly increased green leaf and seed yields in coriander [9,54]. Likewise, the higher yield observed in STCR treatments compared to the general recommended dose can be linked to the balanced nutrient application, which considers the crop requirements and nutrient contributions from soil, fertilizer, and FYM [55]. Additionally, the judicious use of multiple nutrient sources enhances absorption, translocation, and assimilation, ultimately leading to greater dry matter accumulation and improved nutrient content in the crop [56]. STCR-based fertilizer recommendations, particularly those supplemented with FYM, significantly enhance coriander's quality parameters, making them more effective than conventional fertilization approaches. The addition of FYM not only enhanced yield and nutrient uptake but also significantly improved the biochemical quality parameters of coriander, including total phenols, flavonoids, and ascorbic acid, consistent with previous work demonstrating increased antioxidant capacity under organic fertilization programs [57].

The developed equations demonstrated validity as the percent deviation of yield remained within a ± 10 percent range. Targeting yield with only NPK fertilizers resulted in a comparatively higher percentage of achievement than integrated treatments. This was attributed to the greater yields obtained through the inorganic fertilization method [58]. This is supported by research on coriander [6] and chilli [19], where the STCR NPK + FYM strategy outperformed conventional fertilizer recommendations by achieving better nutrient recovery and crop yields. The data also suggests that lower yield targets were more successfully attained than higher ones in cotton [59]. Among the tested targets, the NPK + FYM approach aiming for 10 t ha$^{-1}$ showed a relatively higher RYS compared to its corresponding NPK-alone approach and the 8 t ha$^{-1}$ target, despite achieving significantly higher yields. This could be explained by a more efficient utilization of applied nutrients at lower yield target levels [6]. Similarly, NPK + FYM treatments exhibited higher RYS values compared to their respective NPK-only counterparts. The higher RYS observed under STCR NPK + FYM treatments in contrast to GRD and SFR approaches might be due to a balanced nutrient supply from fertilizers, enhanced utilization of applied fertilizer nutrients in the presence of organic matter, and the beneficial interaction from combining various nutrient sources [60]. Integrated nutrient management has proven effective for coriander cultivation in Alfisols, as studies by [40] and [36] reported that substituting 25% of nitrogen with FYM or applying FYM at 5 t/ha along with the full NPK dose significantly enhanced yield and nutrient uptake. Similar trend of STCR-NPK + FYM outperforming farmer's practice and blanket recommendations was noticed in a maize-tomato cropping sequence [61]. The higher Value Cost Ratio (VCR) under STCR NPK alone for the 10 t ha$^{-1}$ targeted yield could be primarily because of the application of the necessary NPK fertilizer dose without the added cost of FYM, despite the associated higher yields. Although STCR NPK + FYM treatments resulted in greater yields, their VCR was considerably lower, mainly due to the substantial cost of the FYM applied [45].

Partial factor productivity of nitrogen (PFPN) is a robust metric for evaluating nutrient use efficiency (NUE), integrating both the native soil nitrogen contribution and the efficiency with which fertilizer nitrogen is taken up and converted to yield [46]. Typically, PFPN demonstrates a linear decline as nitrogen fertilizer rates increase—an established outcome governed by the law of diminishing returns [62] – and this trend is affirmed by the present findings.For phosphorus, the highest partial factor productivity (PFPP) was recorded in the STCR treatments utilizing a reduced dose of phosphorus fertilizer, supporting efficient phosphorus utilization in coriander [63]. The use of FYM alongside lower chemical fertilizer rates further improved PFPP compared to exclusive NPK treatments, highlighting the beneficial role of organic amendments. FYM enrichment fosters enhanced root development, boosting the soil's nutrient profile and the plants' nutrient absorption capacity [64], thus advancing overall productivity [65].The improved nutrient use efficiency in the STCR approach with lower yield targets may be attributed to the increased yield achieved with reduced fertilizer application rates tailored to crop requirements. Reported values indicate an agronomic efficiency for nitrogen ranging from 6.8 to 34.2 and for potassium between 28.4 and 55.3 in rice crops. The relatively higher agronomic efficiency observed in the STCR-based fertilizer application compared to the general recommended dose may be due to the balanced nutrient supply,

ensuring effective utilization through the combined effects of fertilizers and farmyard manure. Additionally, the relative internal utilization efficiency for nitrogen and phosphorus was significantly higher under STCR-based nutrient management than other fertilizer recommendation methods, while the RIUE for potassium exceeded that of the general recommended dose due to the lower recommended potassium levels.

The STCR approach, with lower fertilizer application rates tailored to crop needs, appears to enhance nutrient use efficiency by optimizing yield. Studies have reported AEN values ranging from 6.8 to 34.2 2 [66] and AEK values between 28.4 and 55.3 in rice cultivation [45]. The relatively higher AE observed with STCR-based fertilizer applications, compared to the general recommended dose, may stem from the balanced nutrient supply and improved utilization facilitated by the synergistic interaction between fertilizers and FYM [6]. Additionally, the IUE for nitrogen and phosphorus is significantly higher under STCR-based nutrient management than other fertilizer recommendation methods, while the IUE for potassium surpasses that of the general recommended dose due to its lower application rates [67].

## Conclusion

In summary, this study demonstrates that the STCR approach, particularly when integrated with farmyard manure, delivers marked benefits over traditional fertilizer recommendation methods in coriander cultivation. STCR enables site-specific, precise nutrient management, resulting in substantial improvements in green foliage yield, nutrient uptake and the biochemical quality of coriander. The superior nutrient use efficiency and economic returns observed in this approach are consistently supported by findings in diverse cropping environments. However, achieving high yields with FYM necessitates considerable investment, and wider implementation is restricted by the availability and cost of organic inputs. The effectiveness of nutrient use and yield enhancements may also fluctuate with variable soil properties, climatic conditions, and management practices, indicating the need for tailored, location-specific calibration. Long-term scalability further depends on validating STCR recommendations across multiple agro-ecological zones and cropping systems. Future advances should focus on refining fertilizer prescription models, incorporating new technologies such as Variable Rate Techniques, and strengthening climate adaptation strategies. Sustainable intensification mandates prioritizing soil health and responsible nutrient stewardship. Broad-based validation, integrating additional organic and biological resources, and examining socio-economic impacts will be crucial for promoting wider adoption.

Overall, the STCR method holds significant promise for boosting the precision, efficiency, and sustainability of coriander production. Its wider potential will be realized through ongoing model improvement, robust validation, and thoughtful integration with local resources and farming needs.

## Supporting information

**S1 Table: Variation in annual rainfall during the field experiments from 2022–2024.**
(DOCX)

**S1 Fig. Layout and details of fertility gradient experiment of fodder maize crop.**
(TIF)

**S2 Fig. The layout and the nutrients level tried under main experiment.**
(TIF)

## Author contributions

**Conceptualization:** Krishna Murthy Rangaiah, Pradip Dey.

**Data curation:** Bhavya Nagaraju.

**Formal analysis:** Bhavya Nagaraju, Shivakumara Maragondanadibba Nanjundappa.

**Funding acquisition:** Sanjay Srivastava, Immanuel Chongboi Haokip.

**Investigation:** Govinda Kasturappa, Shivakumara Maragondanadibba Nanjundappa.

**Methodology:** Govinda Kasturappa.

**Project administration:** Krishna Murthy Rangaiah.

**Resources:** Sanjay Srivastava.

**Supervision:** Krishna Murthy Rangaiah, Sanjay Srivastava, Pradip Dey.

**Validation:** Bhavya Nagaraju.

**Writing – original draft:** Krishna Murthy Rangaiah, Bhavya Nagaraju.

**Writing – review & editing:** Bhavya Nagaraju.

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
