## [Decision Letter · Decision Letter 0]

5 Oct 2025

Dear Dr. Rangaiah,

Thank you for submitting your manuscript to PLOS ONE. After careful consideration, we feel that it has merit but does not fully meet PLOS ONE’s publication criteria as it currently stands. Therefore, we invite you to submit a revised version of the manuscript that addresses the points raised during the review process.

Please consider the enclosed reviewer's comments and make the necessary changes to your manuscript based on their advice/ suggestion and send the revised manuscript for further processing so that we can resend to the reviewers. The authors must provide strong justification, deeper insights on the objectives, research methodology relating to all objectives of the study, soil sampling procedure methodology, statistical analysis, revise the presentation of data and results, perform ANOVA; relevant support on the related crops along with the reasoning in discussion part; conclusion on both strength and limitation along with take home message and future recommendation for the research. Address all the suggestions and comments providing the study data and supportive data. The manuscript cannot be proceeded for further processing if the editors and reviewer’s substantial advice and reservation is not addressed in the revised version of the manuscript.

If you disagree with any comment or comments, kindly state your position and justification.

We look forward to receiving your revised manuscript.

Kind regards,

Nabin Rawal, PhD

Academic Editor

PLOS ONE

Journal Requirements:

"This study was funded by the Indian council of agricultural research and University of

Agricultural Sciences, Bangalore (Grant number: CRP-18)"

3. Please upload a new copy of Figure 4 as the detail is not clear. Please follow the link for more information:  https://journals.plos.org/plosone/s/figures

Additional Editor Comments:

Please consider the enclosed reviewer's comments and make the necessary changes to your manuscript based on their advice/ suggestion and send the revised manuscript for further processing so that we can resend to the reviewers. The authors must provide strong justification, deeper insights on the objectives, research methodology relating to all objectives of the study, soil sampling procedure methodology, statistical analysis, revise the presentation of data and results, perform ANOVA; relevant support on the related crops along with the reasoning in discussion part; conclusion on both strength and limitation along with take home message and future recommendation for the research. Address all the suggestions and comments providing the study data and supportive data. The manuscript cannot be proceeded for further processing if the editors and reviewer’s substantial advice and reservation is not addressed in the revised version of the manuscript.

If you disagree with any comment or comments, kindly state your position and justification.

Reviewers' comments:

Reviewer's Responses to Questions

**Comments to the Author**

1. Is the manuscript technically sound, and do the data support the conclusions?

Reviewer #1: Yes

Reviewer #2: Partly

Reviewer #3: Partly

Reviewer #4: Yes

Reviewer #5: Partly

Reviewer #6: Yes

2. Has the statistical analysis been performed appropriately and rigorously?

Reviewer #1: Yes

Reviewer #2: Yes

Reviewer #3: No

Reviewer #4: No

Reviewer #5: Yes

Reviewer #6: No

3. Have the authors made all data underlying the findings in their manuscript fully available?

Reviewer #1: Yes

Reviewer #2: Yes

Reviewer #3: Yes

Reviewer #4: Yes

Reviewer #5: Yes

Reviewer #6: Yes

4. Is the manuscript presented in an intelligible fashion and written in standard English?

Reviewer #1: Yes

Reviewer #2: Yes

Reviewer #3: No

Reviewer #4: Yes

Reviewer #5: Yes

Reviewer #6: Yes

Reviewer #1: The paper seems to be written well and seems to be a technically sound piece of paper. The experiment has been conducted that meets the protocol for field experiment and lab test using relevant controls. A three year experiments has been utilized which is sufficient to draw the conclusion for coriander crops.

However, after going through the paper i have some comments and concerns before the paper will be published. Please address all the comments I have included in the tract change mode. Some source of data are missing which is very crucial for scientific publications. You have not included enough citation and references for the information you have included in the paper. Provide the methods of the test used for soil and plant sample analysis in tabular format. In the result section for main experiment, please provide the value for coefficient of variation that could provide enough evidence for the stability of the result to draw the conclusion as mentioned in the paper. In the discussion session, please support your results with some literatures from leafy vegetables and spice crops rather then solanaceous crops. Please rewrite your conclusion in simple form readable by scientific community.

Thank you for your experiment and hard work to brings this paper to this form.

Reviewer #2: The manuscript presents a three-year study aimed at developing a Soil Test Crop Response (STCR) model for coriander cultivation in Alfisols. While the topic of precision nutrient management is highly relevant, the manuscript suffers from critical methodological flaws and a lack of conceptual clarity that fundamentally undermine the validity and reliability of the presented results and conclusions. The issues are severe enough that they cannot be addressed through minor or major revisions and necessitate a rejection of the current manuscript. The core problems relate to the experimental design, the derivation and application of the STCR equations, and the interpretation of results.

1. The entire premise of the STCR approach relies on a well-established, consistent fertility gradient. The method of creating this gradient by applying different fertilizer levels to fodder maize is standard. However, the design of the subsequent main experiment with coriander is critically flawed. The manuscript states that within each of the three fertility strips (LFS, MFS, HFS), the 24 treatments (including varying levels of NPK and FYM) were randomized. This means that the very nutrient applications that are the treatments in the main experiment (NPK and FYM) are being applied on top of the pre-existing, drastically different fertility levels. This creates an uncontrollable and unquantifiable interaction. The response of coriander to, for example, 35 kg N/ha will be entirely different when applied to a Low Fertility Strip versus a High Fertility Strip. The basic parameters (NR, CS, CF) calculated from this experiment are therefore confounded by the initial fertility gradient, making them invalid for developing universal equations. A valid STCR approach would have conducted the main experiment on a uniform field, not one with an pre-imposed gradient.

2. The equations presented (EQ-1 to EQ-6) are the central output of the study. However, their derivation is not clearly explained from the collected data. The manuscript jumps from describing the calculation of basic parameters to presenting the final equations without detailing the statistical model (e.g., multiple regression analysis using yield as a function of soil test values and nutrient doses) that is standard for STCR development. The provided multivariate regression model (Table 4) uses only soil test values (SN, SP, SK) to predict yield (Y), which is insufficient for fertilizer prescription; it completely omits the applied fertilizer doses, which are the critical variables for a prescription model. This suggests a fundamental misunderstanding of the STCR methodology.

3. The data in Table 6 and Table 7, which compare the validation trial treatments, lack the necessary statistical analysis. While SEm and CD are provided for Table 6, there is no ANOVA table showing the significance of the overall model. More critically, the means in the tables are not accompanied by letters indicating significant differences. It is therefore impossible for the reader to determine if the reported differences between STCR, GRD, and SFR treatments (e.g., "56.90% higher yield") are statistically significant or merely numerical variations.

The correlation analyses in Figures 5, 6, and 7 are poorly presented. The R² values are illegible in the provided figures, and the narrative describing these correlations is vague and contradictory (e.g., "significant linear correlation" vs. "low correlation... showed its nonsignificant effect").

Reviewer #3: The manuscript entitled "Viable Fertilizer Prescription Model: Soil Test Crop Response Approach for Sustained and Targeted Yield, Quality of Coriander (Coriandrum sativum L.) in Alfisols" focuses on formulating precise fertilizer recommendations that optimize nutrient uptake and use to improve coriander productivity in India. Although the research topic is not highly innovative, the authors have addressed the problem rigorously and presented interesting data. However, major revisions are required before the manuscript can be considered for possible publication.

Abstract:

- This section should be rewritten more clearly. The experiments should be better explained, and all abbreviations should be explicitly defined.

Introduction:

- Include more references on the Soil Test Crop Response (STCR) method, citing studies by other authors who have applied this approach (for example: https://doi.org/10.3390/agronomy11091756 or https://doi.org/10.3390/su14148629), and also mention alternative methods such as Variable Rate Techniques (VRT) (https://doi.org/10.1371/journal.pone.0267219).

- Lines 78–84: the information provided is already well known and could be removed.

Materials and Methods:

- Since this study is based on soil baseline characterization, further details are needed. How many samples were analyzed? When were they collected? Was soil pH measured in aqueous extract? What are the percentages of the textural classes?

- In the paragraph "Chemical analysis of soil and plant samples" soil analysis is described again, as it is in "Details of the experimental field." It would be better to harmonize the presentation of analytical methods and avoid repeating them in two different sections.

- In the paragraph "Quality parameters of coriander", why were agronomic parameters not considered?

Results:

- Line 249: remove the word “also.”

- When presenting results, statistical analyses should be explicitly referenced.

Discussion:

- This section should be more concise; results should not be repeated but only interpreted.

- The first part of the discussion is rather obvious and could be shortened.

- More references to similar studies should be included.

Conclusions:

Given the work carried out, the conclusions should focus on both the strengths and limitations of the STCR method, and propose future perspectives aimed at improving the approach.

Additional Comments:

- The quality of the figures is poor and should be improved.

- The English language requires revision.

Reviewer #4: Summary

The study develops Soil Test Crop Response (STCR) equations (with/without FYM) to prescribe N-P-K for coriander on Alfisols and validates them against GRD and SFR in a randomized design. The topic is important for site-specific fertilizer recommendations, and the ready-reckoner has practical value. However, several major issues must be addressed to ensure rigor, clarity, and reproducibility.

Major Issues

1. The description of the experimental field does not include a summary of seasonal weather conditions (lines 92 – 107). Authors should include a summary of seasonal weather conditions.

2. Line 156: “Table 1: Treatment structure for the main test crop experiment of maize” conflicts with the text (test crop experiment with coriander in line 126). Please change “maize” to “coriander.”.

3. In the FYM-integrated formulas (Line 164), the FK expression subtracts terms involving SP (P) rather than SK (K). Please correct to SK consistently.

4. Authors should include the following missing operational details in the materials and methods section.

i) plot size dimensions and plant spacing for the fertility gradient experiment (line 110) and the test crop experiment (line 125)

ii) analysis and handling of the FYM

iii) any other management practices (pest or weed management)

5. Provide ANOVA/GLM details for the main and validation experiments, including model structure (blocks, strips, FYM levels), multiple-comparison procedure (e.g., Tukey/LSD with α), diagnostics (normality/homoscedasticity), and exact P values. Regression adequacy should include residual plots and collinearity checks. Consider prediction performance for the validation (e.g., bias, RMSE, MAPE between targeted and achieved yield). The current regression summary alone is insufficient to support broad performance claims.

6. From lines 339 to 341, the authors reported that when the ready-reckoner produces soil N test values above 480 kg ha-1, the fertilizer nutrients required for the same yield target are negative (line 346) but then advised applying 25–30% of the “recommended dose” to maintain soil fertility status. The authors should clarify the agronomic rationale or justification of that statement and provide reference.

7. In Discussion, explicitly state what is novel relative to prior STCR work in spices/leafy vegetables (coefficient magnitudes, FYM integration, operationalization via ready-reckoner) and clarify limits of transferability.

Minor Issues

1. “soil test crop response-based” should be hyphenated consistently.

2. Ensure all abbreviations appear once and in order; remove duplicates.

3. Check for grammatical and typological erros.

4. Use consistent units (e.g., “kg ha⁻¹” and “t ha⁻¹”) throughout the manuscript.

Reviewer #5: i) The authors reported that the soils of the experimental site are classified as Typic Kandic Paleustalfs with a soil family of fine, mixed, isohyperthermic. They further described the soils as well-drained, deep, red sandy loam. This description creates confusion regarding both the textural conditions and the drainage status of the study site. Since the soils possess a kandic horizon, they are characterized by low-activity clays with poor nutrient- and water-holding capacity. Furthermore, coriander generally grows well in soils with a pH range of 6.0–7.0, yet they reported pH of the site is 6.0, making it acidic and therefore limiting nutrient availability. This raises an important question: why did the authors select such inherently poor-quality soils to develop a viable fertilizer prescription model? In fact, these soils are typically more suitable for plantation crops rather than spice crops such as coriander.

ii) Regarding the fertility gradient history (Phase I), the authors indicated that they selected a field with uniform characteristics and imposed differential fertilizer treatments using a high nutrient-responsive exhaust crop, which is indeed a prerequisite for such experiments. However, another critical requirement is allowing adequate time for natural processes, including plant and microbial interactions. The authors failed to specify how many days were allocated for root–microbe interactions to occur. In addition, they did not provide any details about the previous cropping history of the experimental site or its management practices. Without this information, the fertility gradient history does not present a complete or accurate picture of the experiment.

iii) The authors also did not adequately explain the soil sampling procedure. They merely mentioned that "standard procedures" were followed, which is an incomplete and insufficient statement for a well-documented scientific study. Similarly, in Phase II, the methodology for soil sampling is missing, which undermines the rigor of the experiment.

Reviewer #6: General Comments

The manuscript is interesting and provides valuable insights into Soil Test Crop Response. However, there are several concerns that should be addressed before it can be considered further.

1. Treatment Setup

The treatment arrangement for maize is not clearly described and may not be replicable.

It is not clear which treatment is not designated as Treatment 1, nor is the total number of treatments specified. Please provide a clear description of the treatment structure.

2. Statistical Analysis

The results are presented using only simple statistics. This approach is not sufficient to support the conclusions.

Why was ANOVA not performed? Please analyze the data using ANOVA and include the corresponding ANOVA table showing sources of variation, degrees of freedom, mean squares, F-values, and significance levels.

The response of each treatment is not clearly presented. Currently, the table only shows treatment groups as LFS, MFS, and HFS. Presenting the results in ranges makes it less understandable. (Table 2).

It would be more appropriate to show the treatment effects explicitly based on the nutrient application levels (e.g., 0, 1, 2, 3, etc.), rather than only grouping them as LFS, MFS, and HFS. This will make the treatment comparisons clearer and more informative.

3. Editorial Issues

There are some editorial errors that should be corrected during revision.

In the abstract, capitalization is used inconsistently, which affects readability. Please revise for consistency and clarity.

**Do you want your identity to be public for this peer review?** For information about this choice, including consent withdrawal, please see our Privacy Policy

Reviewer #1: **Yes:** Madan Marasini

Reviewer #2: **Yes:** Muhammad Usman

Reviewer #3: No

Reviewer #4: No

Reviewer #5: No

Reviewer #6: **Yes:** Ewunetie Melak

---

## [Author Response · Author response to Decision Letter 1]

29 Oct 2025

Reviewer #1:

1. Some source of data are missing which is very crucial for scientific publications.

Manuscript is carefully reviewed and included

2. You have not included enough citation and references for the information you have included in the paper.

Corrected and additional references were included

3. Provide the methods of the test used for soil and plant sample analysis in tabular format.

Soil and plant analysis tests were presented separately in a tabular format.

4. In the result section for main experiment, please provide the value for coefficient of variation that could provide enough evidence for the stability of the result to draw the conclusion as mentioned in the paper.

The coefficient of variation values were included in the respective tables.

5. In the discussion session, please support your results with some literatures from leafy vegetables and spice crops rather then solanaceous crops.

Included

6. Please rewrite your conclusion in simple form readable by scientific community.

Corrected

Reviewer #2:

1. The entire premise of the STCR approach relies on a well-established, consistent fertility gradient. The method of creating this gradient by applying different fertilizer levels to fodder maize is standard. However, the design of the subsequent main experiment with coriander is critically flawed. The manuscript states that within each of the three fertility strips (LFS, MFS, HFS), the 24 treatments (including varying levels of NPK and FYM) were randomized. This means that the very nutrient applications that are the treatments in the main experiment (NPK and FYM) are being applied on top of the pre-existing, drastically different fertility levels. This creates an uncontrollable and unquantifiable interaction. The response of coriander to, for example, 35 kg N/ha will be entirely different when applied to a Low Fertility Strip versus a High Fertility Strip. The basic parameters (NR, CS, CF) calculated from this experiment are therefore confounded by the initial fertility gradient, making them invalid for developing universal equations. A valid STCR approach would have conducted the main experiment on a uniform field, not one with an pre-imposed gradient.

The STCR methodology used in our study is based on the established fixed plot design by Ramamoorthy et al. (1967), which deliberately creates a fertility gradient using a preliminary crop (fodder maize) to represent varied field fertility conditions. The intention is to assess crop response to fertilizer across a realistic spectrum of soil fertility. By randomizing treatments within each fertility strip, the experiment captures how different NPK and FYM combinations interact with specific fertility levels—this is a key feature, not a flaw, of the STCR approach, allowing calculation of nutrient requirements and contributions from soil, fertilizer, and organic sources tailored to varying field conditions. Validation trials were subsequently carried out on uniform fields to confirm the consistency and applicability of the developed fertilizer equations. Thus, our design follows standard STCR methodology and ensures robust, site-specific fertilizer recommendations for coriander.

2. The equations presented (EQ-1 to EQ-6) are the central output of the study. However, their derivation is not clearly explained from the collected data. The manuscript jumps from describing the calculation of basic parameters to presenting the final equations without detailing the statistical model (e.g., multiple regression analysis using yield as a function of soil test values and nutrient doses) that is standard for STCR development. The provided multivariate regression model (Table 4) uses only soil test values (SN, SP, SK) to predict yield (Y), which is insufficient for fertilizer prescription; it completely omits the applied fertilizer doses, which are the critical variables for a prescription model. This suggests a fundamental misunderstanding of the STCR methodology.

The equations (EQ-1 to EQ-6) were developed using the classical STCR approach of Ramamoorthy et al. (1967), integrating soil test values, target yields, and nutrient doses to create fertilizer prescription models. Key parameters—nutrient requirement (NR), and nutrient contributions from soil (CS), fertilizer (CF), and FYM (CFYM)—were calculated from yield and nutrient uptake data. Unlike the multivariate regression model in Table 4 that only uses soil test nutrients to predict yield, the actual STCR fertilizer prescription equations incorporate soil test values, target yield, and organic manure levels to calculate precise fertilizer doses . Validation trials showed these equations improve coriander yield and nutrient use efficiency, highlighting their ability to balance soil, fertilizer, and organic inputs for specific yield goals in line with the established STCR methodology.

3. The data in Table 6 and Table 7, which compare the validation trial treatments, lack the necessary statistical analysis. While SEm and CD are provided for Table 6, there is no ANOVA table showing the significance of the overall model. More critically, the means in the tables are not accompanied by letters indicating significant differences. It is therefore impossible for the reader to determine if the reported differences between STCR, GRD, and SFR treatments (e.g., "56.90% higher yield") are statistically significant or merely numerical variations. The correlation analyses in Figures 5, 6, and 7 are poorly presented. The R² values are illegible in the provided figures, and the narrative describing these correlations is vague and contradictory (e.g., "significant linear correlation" vs. "low correlation... showed its nonsignificant effect").

Letters indicating statistically significant differences, as determined by Tukey’s HSD test at the 5% probability level, have now been added to each treatment mean in the Table 6. For Table 7, ANOVA was not performed on parameters like nutrient use efficiency indices (PFP, AE, PNB, IUE) because these values are calculated for each treatment based on aggregated yield and nutrient input data, resulting in a single value per treatment without replication. The R² values in all figures have now been enhanced for improved clarity and legibility. The narrative description has been revised for consistency and scientific accuracy, avoiding contradictory statements. We have ensured that the terms "significant" and "non-significant" are used strictly based on statistical results and relevant p-values. Please refer to the updated table for specific corrections and clarifications.

Reviewer 3

Abstract:

- This section should be rewritten more clearly. The experiments should be better explained, and all abbreviations should be explicitly defined.

Rewritten and included

Introduction:

- Include more references on the Soil Test Crop Response (STCR) method, citing studies by other authors who have applied this approach (for example: https://doi.org/10.3390/agronomy11091756 or https://doi.org/10.3390/su14148629), and also mention alternative methods such as Variable Rate Techniques (VRT) (https://doi.org/10.1371/journal.pone.0267219).

Included in the introduction part

- Lines 78–84: the information provided is already well known and could be removed.

The background / role of nutrients are required in the introduction part to justify the importance of considering NPK for development of targeted yield equations.

Materials and Methods:

- Since this study is based on soil baseline characterization, further details are needed. How many samples were analyzed? When were they collected?

The specifics regarding the quantity and timing of soil and plant sample collection were provided

Was soil pH measured in aqueous extract? What are the percentages of the textural classes?

Yes, initial soil pH was measured in an aqueous extract (pH 5.98, 1:2.5 soil: water ratio following Jackson, 1973). The site was identified as deep red sandy loam, with its texture confirmed sandy loam by the International Pipette method. Detailed analysis revealed proportions of sand, silt, and clay were 63.12%, 12.55%, and 24.33%, respectively—an optimal composition for coriander cultivation.

- In the paragraph "Chemical analysis of soil and plant samples" soil analysis is described again, as it is in "Details of the experimental field." It would be better to harmonize the presentation of analytical methods and avoid repeating them in two different sections.

Corrected and analytical procedures for soil and plant nutrient estimation were kept in seperate tables to avoid redundancy.

- In the paragraph "Quality parameters of coriander", why were agronomic parameters not considered?

The quality parameter section focused on key secondary metabolites—total phenols, flavonoids, and ascorbic acid—as nutritional indicators of coriander foliage. Agronomic parameters such as plant height, biomass, and yield were analyzed within the yield and nutrient uptake sections. This structure follows STCR literature, presenting agronomic data with yield efficiency and discussing quality traits separately to emphasize biochemical improvements.

Results:

- Line 249: remove the word “also.”

Deleted

- When presenting results, statistical analyses should be explicitly referenced.

Statistical analyses are now explicitly referenced at the beginning of the results section

Discussion:

- This section should be more concise; results should not be repeated but only interpreted.

The Discussion has been substantially revised for brevity and interpretative clarity. Detailed numerical results and experimental outcomes are no longer repeated.

- The first part of the discussion is rather obvious and could be shortened.

The introductory portion of the Discussion was condensed, omitting general statements about balanced fertilization and shifting to a direct evaluation of site-specific nutrient management outcomes realized through the STCR approach in coriander.

- More references to similar studies should be included.

Included multiple references to recent and relevant research

Conclusions:

Given the work carried out, the conclusions should focus on both the strengths and limitations of the STCR method, and propose future perspectives aimed at improving the approach.

Corrections are included in the manuscript

Additional Comments:

- The quality of the figures is poor and should be improved.

Corrected

- The English language requires revision.

Corrected

Reviewer #4: Summary

Major Issues

1. The description of the experimental field does not include a summary of seasonal weather conditions (lines 92 – 107). Authors should include a summary of seasonal weather conditions.

Details of the seasonal weather conditions are presented in Supplementary Table 1

2. Line 156: “Table 1: Treatment structure for the main test crop experiment of maize” conflicts with the text (test crop experiment with coriander in line 126). Please change “maize” to “coriander.”.

Checked and corrected

3. In the FYM-integrated formulas (Line 164), the FK expression subtracts terms involving SP (P) rather than SK (K). Please correct to SK consistently.

Checked and corrected

4. Authors should include the following missing operational details in the materials and methods section.

i) plot size dimensions and plant spacing for the fertility gradient experiment (line 110) and the test crop experiment (line 125)

Fertility gradient: Plot size 8.1 m X 2.7 m Plant spacing: 30 cm between the plants x 60 cm between the rows for fodder maize

Main crop experiment: Plot size 8.1 m X 2.7m Plant spacing: 22.5 cm x10 cm for coriander

ii) analysis and handling of the FYM

Farmyard manure was incorporated into the soil 15 days before sowing the main crop, following the recommended application rate of 7.5 t ha⁻¹. The analysis of FYM was conducted following standard analytical protocols (Jackson, M.L., 1973), and the results indicated nutrient contents of 0.51 % nitrogen, 0.28 % phosphorus, and 0.47 % potassium.

iii) any other management practices (pest or weed management)

Details of weed and pest control (herbicide/insecticide usage, manual interventions, timing) will be provided as per UAS Bangalore agronomic package of practices.

5. Provide ANOVA/GLM details for the main and validation experiments, including model structure (blocks, strips, FYM levels), multiple-comparison procedure (e.g., Tukey/LSD with α), diagnostics (normality/homoscedasticity), and exact P values. Regression adequacy should include residual plots and collinearity checks. Consider prediction performance for the validation (e.g., bias, RMSE, MAPE between targeted and achieved yield). The current regression summary alone is insufficient to support broad performance claims.

The main experiments were summarized using descriptive statistics. Data from validation trials were analyzed through ANOVA using OPSTAT and SPSS v16.0, and mean comparisons were conducted using the least significant difference (LSD) test at α = 0.05. Model assumptions were verified through the Shapiro–Wilk and Levene’s tests, while regression diagnostics involved residual analysis and multicollinearity checks using variance inflation factors (VIF) and correlation matrices. Model performance was evaluated based on bias, root mean square error (RMSE), and mean absolute percentage error (MAPE) to determine yield prediction accuracy. Pearson correlation analysis (using Excel) was applied to assess relationships among yield, nutrient uptake, and soil organic carbon. Multivariate regression outputs included p-values and confidence intervals.

6. From lines 339 to 341, the authors reported that when the ready-reckoner produces soil N test values above 480 kg ha-1, the fertilizer nutrients required for the same yield target are negative (line 346) but then advised applying 25–30% of the “recommended dose” to maintain soil fertility status. The authors should clarify the agronomic rationale or justification of that statement and provide reference.

The recommendation to apply 25–30% of the "recommended dose" of fertilizers, even when STCR-based calculations yield negative values, is to prevent soil nutrient depletion and maintain long-term soil fertility. Negative doses indicate sufficient nutrient availability for the targeted yield, but completely withholding fertilizers risks gradual nutrient mining from the soil. Applying this minimal dose helps replenish nutrients lost through crop uptake and leaching, supports microbial activity, and preserves soil health for future crops. The STCR approach balances precise fertilizer use for yield targets while ensuring sustainable nutrient supply by maintaining baseline fertilizer inputs. This guidance aligns with STCR research emphasizing nutrient balance for sustained productivity and soil fertility conservation (Saxena et al., 2008; Dey et al., 2015 and All India Coordinated STCR reports).

7. In Discussion, explicitly state what is novel relative to prior STCR work in spices/leafy vegetables (coefficient magnitudes, FYM integration, operationalization via ready-reckoner) and clarify limits of transferability).

Included

Minor Issues

1. “soil test crop response-based” should be hyphenated consistently.

Corrected

2. Ensure all abbreviations appear once and in order; remove duplicates.

Included

3. Check for grammatical and typological erros.

Checked and corrected

4. Use consistent units (e.g., “kg ha⁻¹” and “t ha⁻¹”) throughout the manuscript.

Corrected

Reviewer #5:

i) The authors reported that the soils of the experimental site are classified as Typic Kandic Paleustalfs with a soil family of fine, mixed, isohyperthermic. They further described the soils as well-drained, deep, red sandy loam. This description creates confusion regarding both the textural conditions and the drainage status of the study site. Since the soils possess a kandic horizon, they are characterized by low-activity clays with poor nutrient- and water-holding capacity. Furthermore, coriander generally grows well in soils with a pH range of 6.0–7.0, yet they reported pH of the site is 6.0, making it acidic and therefore limiting nutrient availability. This raises an important question: why did the authors select such inheren

---

## [Decision Letter · Decision Letter 1]

12 Jan 2026

Viable Fertilizer Prescription Model:  Soil Test Crop Response Approach for Sustained and Targeted Yield, Quality of Coriander (Coriandrum sativum L.) in Alfisols

PONE-D-25-26444R1

Dear Dr. Rangaiah,

We’re pleased to inform you that your manuscript has been judged scientifically suitable for publication and will be formally accepted for publication once it meets all outstanding technical requirements.

Kind regards,

Nabin Rawal, PhD

Academic Editor

PLOS One

Additional Editor Comments (optional):

We truly appreciate your consideration to publish your research findings with the PLOS One. Thank you for addressing the comments and suggestion from subject matter experts (all reviewers), I would like to inform you that it may be proceeded for further processing. Please check all the journal formating and references during final proof.

Reviewers' comments:

Reviewer's Responses to Questions

**Comments to the Author**

Reviewer #1: All comments have been addressed

Reviewer #2: All comments have been addressed

Reviewer #5: All comments have been addressed

Reviewer #6: All comments have been addressed

2. Is the manuscript technically sound, and do the data support the conclusions?

Reviewer #1: Yes

Reviewer #2: Yes

Reviewer #5: Yes

Reviewer #6: Partly

3. Has the statistical analysis been performed appropriately and rigorously?

Reviewer #1: Yes

Reviewer #2: Yes

Reviewer #5: Yes

Reviewer #6: N/A

4. Have the authors made all data underlying the findings in their manuscript fully available?

Reviewer #1: Yes

Reviewer #2: Yes

Reviewer #5: Yes

Reviewer #6: Yes

5. Is the manuscript presented in an intelligible fashion and written in standard English?

Reviewer #1: Yes

Reviewer #2: Yes

Reviewer #5: Yes

Reviewer #6: Yes

Reviewer #1: The author has adressed all the feedback provided in the previous review. I think the paper is now in correct fotm for publication as it represent the validation experiment as well as field research work to support the conclusion.

Reviewer #2: Authors have done all suggestions and comments. I recomend the MS for publication in PLOS One journal.

Reviewer #5: Authors are advised to follow the journal’s prescribed style and formatting guidelines. They should provide up-to-date references to support their findings and ensure that all sources are properly cited. Authors must also confirm that their work is original, previously unpublished, and free from plagiarism, data fabrication, or data manipulation.

Reviewer #6: I recommended to be published,but some editorial error should be addressed before publication and the limitation of the study should be stated on the conclusion and recommendation part.

**Do you want your identity to be public for this peer review?** For information about this choice, including consent withdrawal, please see our Privacy Policy

Reviewer #1: No

Reviewer #2: No

Reviewer #5: **Yes:** Md. Jashim Uddin

Reviewer #6: No

---

## [Editor Report · Acceptance letter]

PONE-D-25-26444R1

PLOS One

Dear Dr. Rangaiah,

I'm pleased to inform you that your manuscript has been deemed suitable for publication in PLOS One. Congratulations! Your manuscript is now being handed over to our production team.

Kind regards,

on behalf of

Dr. Nabin Rawal

Academic Editor

PLOS One